# Evolutionary algorithm using surrogate models for solving bilevel multiobjective programming problems

Yuhui Liu[1]◉, Hecheng Li[2]◉*, Hong Li[3]

**1** School of Computer Science and Technology, Qinghai Normal University, Xining, China, **2** School of Mathematics and Statistics, Qinghai Normal University, Xining, China, **3** School of Mathematics and Statistics, Xidian University, Xi'an, China

◉ These authors contributed equally to this work.
* lihecheng@qhnu.edu.cn

**Data Availability Statement:** All relevant data are within the manuscript and its Supporting information files, and I also uploaded data on URL https://doi.org/10.5061/dryad.dfn2z3504.

## Abstract

A bilevel programming problem with multiple objectives at the leader's and/or follower's levels, known as a bilevel multiobjective programming problem (BMPP), is extraordinarily hard as this problem accumulates the computational complexity of both hierarchical structures and multiobjective optimisation. As a strongly NP-hard problem, the BMPP incurs a significant computational cost in obtaining non-dominated solutions at both levels, and few studies have addressed this issue. In this study, an evolutionary algorithm is developed using surrogate optimisation models to solve such problems. First, a dynamic weighted sum method is adopted to address the follower's multiple objective cases, in which the follower's problem is categorised into several single-objective ones. Next, for each the leader's variable values, the optimal solutions to the transformed follower's programs can be approximated by adaptively improved surrogate models instead of solving the follower's problems. Finally, these techniques are embedded in MOEA/D, by which the leader's non-dominated solutions can be obtained. In addition, a heuristic crossover operator is designed using gradient information in the evolutionary procedure. The proposed algorithm is executed on some computational examples including linear and nonlinear cases, and the simulation results demonstrate the efficiency of the approach.

## Introduction

### Problem models

The bilevel programming problem (BLPP), a hierarchical optimisation problem, has a nested optimisation structure because of a leader and a follower. In a BLPP, optimisation procedures are executed successively at both the leader's and follower's levels. The upper-level optimisation is provided by the leader, whereas the lower-level optimisation is controlled by the follower. The related problems are called the upper-level/leader's problem and lower-level/follower's problem. Both the leader's and follower's problems have their own decision

**Funding:** The research work was supported by the National Natural Science Foundation of China under Grant Nos. 61966030 and 11661067, the Natural Science Foundation of Qinghai Province under Grant No. 2018-ZJ-901 and the Key Laboratory of IoT of Qinghai under grant 2020-ZJ-Y16.

**Competing interests:** The authors have declared that no competing interests exist.

variables, constraints and objective functions [1]. The general BLPP can be formulated as follows:

$$
\begin{cases}
\min_{x} \tilde{F}(x, y) \\
s.t. G_k(x, y) \le 0, k = 1, 2, \cdots, K \\
\min_{y} \tilde{f}(x, y) \\
s.t. g_j(x, y) \le 0, j = 1, 2, \cdots, J
\end{cases}
\tag{1}
$$

Where $x = (x_1, \ldots, x_n)$ are the leader's variables, $y = (y_1, \ldots, y_m)$ the follower's variables, $\tilde{F} : R^{n+m} \rightarrow R$ the leader's objective function, $\tilde{f} : R^{n+m} \rightarrow R$ the follower's objective function, $G_k: R^{n+m} \rightarrow R, k = 1, 2, \cdots, K$, the leader's constraints, and $g_j: R^{n+m} \rightarrow R, j = 1, 2, \cdots, J$, the follower's constraints.

When the number of objectives in a leader's and/or follower's problems exceeds one, the problem is called a bilevel multiobjective programming problem (BMPP). The BMPP is one of the most difficult optimisation problems, because it accumulates all computational costs of hierarchical and multiobjective optimisations. Consequently, it is strongly NP-hard. The BMPP can be formulated as follows:

$$
\begin{cases}
\min_{x} F(x, y) = (F_1(x, y), F_2(x, y), \ldots, F_q(x, y)) \\
s.t. G_k(x, y) \le 0, k = 1, 2, \cdots, K \\
\min_{y} f(x, y) = (f_1(x, y), f_2(x, y), \ldots, f_p(x, y)) \\
s.t. g_j(x, y) \le 0, j = 1, 2, \cdots, J
\end{cases}
\tag{2}
$$

Here, $a \le x \le b$, where $a$, and $b$ are the lower and upper bounds of $x$, respectively. In general, the optimisation procedure of problem (2) can be described as follows. The leader first selects a strategy $x$ to optimise his objective, and then the follower reacts to the leader's selection by providing a group of non-dominated solutions $y_i$ to the follower's problem. The pairs $(x, y_i)$ are then used as bilevel feasible solutions to (2). When the leader selects another strategy, the related feasible point pairs can be obtained. To solve (2) is analogous to determining a set of non-dominated solutions that is distributed well for the leader's objective in all bilevel feasible solutions.

Unlike BLPPs with a single objective at each level, the BMPP incurs an additional computational cost for the selection of the follower's non-dominated solutions. In addition, the selection of Pareto solutions in the leader's problem renders the BMPP much harder than the single objective case.

## Related work

For problem (1), BLPPs have been extensively applied in economic management and engineering, urban traffic and transportation, supply chain planning, resource assignment, engineering design, structural optimisation, and game-playing strategies. For example, Zhu and Guo [2] proposed a BLPP with a maxmin or minmax operator in the follower's levels for a manufacturer, where the manufacturer plans to produce multiple short life cycle products using one-shot decision theory. The model was solved using typically used optimisation methods. Based on a bilevel complementary model, Nasrolahpouret et al. [3] developed an energy storage system decision tool for merchant price-making, that can determine the most

advantageous trading behaviour in a pool-based market. More examples in real-world applications are summarised in [4–8].

As the applications of BLPPs are becoming increasingly extensive and diverse, researchers have focused on developing efficient solution strategies for such problems. However, owing to the intrinsic non-convexity and non-differentiability of BLPPs, a general BLPP is always complex and extremely challenging to solve using canonical optimisation methods that involve gradient information. Thus far, only a limited number of BLPPs can be solved effectively, such as linear [9–13] and convex quadratic BLPPs [14–16]. Liu et al. [10] used a new variant of the penalty method and Karush-Kuhn-Tucker(K-K-T) conditions to solve weak linear BLPPs. Franke et al. [16] used K-K-T conditions and the optimal value approach to transform a bilevel convex programming problem into a single-level optimisation problem, and introduced M-stationarity for mathematical programs with complementarities in Banach spaces. For most general BLPPs, only local optima can be obtained using these gradient approaches. To effectively solve BLPPs, another algorithm framework, i.e. the swarm optimisation method, has been widely adopted. This method is based on population search technology and affords good global convergence. Evolutionary algorithms [17–19], as representative swarm optimisation techniques, have been widely adopted to develop various bilevel optimisation algorithms in the past decades. For example, Sinha et al. [19] presented a single-level reduction of BLPPs using approximate K-K-T conditions and embedded the neighbourhood measure idea into an evolutionary algorithm. Based on a new constraint-handling scheme, Wang, et al. [20] proposed an evolutionary algorithm using K-K-T conditions. In Wang's method, the new constraint-handling scheme enables individuals to satisfy linear constraints.

For problem (2), even though it is extremely difficult to design efficient solution approaches for the BMPP, a large number of practical applications stimulate the researches on theoretical results and algorithmic design. Guo and Xu [21] developed a BMPP model to study the seismic risk of transportation system reconstruction in large construction projects, and fuzzy random variable transformation and fuzzy variable decomposition methods were proposed to solve the model. Brian et al. [22] proposed a BMPP model for coordinating multiple design problems according to conflicting criteria. The design of a hybrid vehicle layout was expressed as a two-stage decomposition problem including vehicle class and battery class, and a multiobjective decomposition algorithm was developed. Alves et al. [23] developed a particle swarm optimisation algorithm to solve linear BMPPs with simple multiple objectives at the leader's level. Chakraborti et al. [24] investigated environmental-economic power generation and despatch problems using the BMPP.

The existing BMPPs can be classified into three categories: 1) Only the leader's problem involves multiobjective optimisation [25, 26]; 2) only the follower's problem includes multiobjective optimisation [27–29]; and 3) both levels involve multiobjective optimisation [30–39]. In fact, once a the non-dominated procedure is hierarchically executed, the computational amount for bilevel optimisation will be significant. Consequently, a few efficient approaches exist for addressing linear or nonlinear BMPPs.

For linear BMPPs, Calice et al. [34] first converted the linear BMPP into two multiobjective problems, and then used some specific optimisation conditions to prove that the common solutions to each multiobjective problem are optimal to the original problem. Kirlik et al. [35] proposed the use of a global optimisation method for discrete linear BMPPs. However, this method is computationally intensive. Liu et al. [36] investigated a class of pessimistic semi-rectorial BLPPs, and used secularisation to, transform the original problem into a scalar objective optimisation problem. Subsequently, the authors used the generalised differentiation calculus of Mordukhovich to establish optimality conditions for linear multiobjective problems. Lv and Wan [37] used the duality gap as a penalty on the leader's objective, and the follower's problem

was transformed into a single objective case by adopting a weighted sum scheme. Alves [25] used a multiobjective particle swarm optimisation algorithm to solve linear BMPPs with multiple objective functions at the leader's level.

Only a few studies have been reported regarding nonlinear BMPPs. Deb and Sinha [29] presented an evolutionary-local-search algorithm for solving nonlinear BMPPs. In that study, the mapping method was used for the first time to solve the follower's programming problems. Jia et al. [30] used genetic algorithms to solve BMPPs, in which a uniform design was used to generate weights, crossover and mutation operators. In addition, the follower's objectives were converted into a single objective function by the weight sum method. In this method, MOEA/D was used to solve the leader's multiobjective convex programming problem. Besides, based on the sensitivity theorem, an iterative method was used to solve a class of nonlinear BMPPs [33]. Zhang et al. [38] presented an improved particle swarm optimisation for solving BMPPs.

## Research motivation

As mentioned above, because BMPPs have high computational complexity, only a few efficient solution methods exist for general BMPPs, and most of the existing approaches have been developed to solve cases where only the leader's problem is multiobjective. When the follower's problem involves multiple objectives, the non-dominated procedure can cause a large amount of computation. In addition, recall that in BLPPs, the optimisation procedure of the follower's problem is frequently executed to obtain bilevel feasible points, which can increase the computational cost. Consequently, if one intends to develop an efficient approach, two key issues must be addressed. One is to reduce the computational cost caused by the follower's optimisation procedure, whereas the other is to avoid excessive dominant comparisons among individuals. Hence, the weighted sum method is always adopted to delete non-dominated sorting procedures [40], and K-K-T conditions are applied to transform BLPPs into single-level ones. Furthermore, either MOEA/D [41, 42] or NSGA-II [43, 44] can be used to solve multiobjective optimisation problems. However, weight sum and K-K-T transformation techniques cannot perform the computation effectively when addressing the follower's problems. In the literature [26] Li et al used evolutionary algorithms to solve linear BMPPs in which the leader's objective is multiobjective. In Li's method, the programming problem was first transformed into a single-level multiobjective problem by K-K-T conditions; subsequently. The multiobjective problem was solved via the weighted sum method, Tchebycheff approach and NSGA-II method, separately. Finally, the computational results obtained using the three approaches were compared. Sinha et al. [31] presented an approximation of the set-mapping method to solve the follower's problem. This method can effectively reduce the amount of computation caused by the follower's optimisation. However, it is complicated to establish multiple quadratic fibers between the variables of the leader's and follower's problems. The methods used in the literature [26] and [31] requires a high computational cost.

Motivated by both weighted aggregation and approximate solution methods, in this study, an evolutionary algorithm using surrogate models was developed to solve BMPPs. Unlike the existing algorithms, the proposed algorithm is characterised as follows. First, a weighted aggregation method using a uniform design is adopted to convert the follower's problem into several single objective ones. Second, surrogate optimisation models and interpolation functions are used to replace the solution functions of the follower's problem and updated periodically using new sample points. Finally, as heuristic information, the gradient direction is embedded into genetic operators to produce potentially high-quality offspring.

## Basic notions

Some basic definitions for problem (2) are summarized as follows:

Let $B_1 = \{(x, y)|G_k(x, y) \leq 0, x \in R^n, y \in R^m, k = 1, 2, \ldots, K\}$ and $B_2 = \{(x, y)|g_j(x, y) \leq 0, x \in R^n, y \in R^m, j = 1, 2, \ldots, J\}$.

**Definition 0.1** (*Dominated relations*): *Vector* $F^1 = (F_1^1, \ldots, F_1^q)$ *is dominated by vector* $F^2 = (F_2^1, \ldots, F_2^q)$, *denoted by* $F^2 \prec F^1$, *if* $F_2^i \leq F_1^i, i = 1, 2, \ldots, q$, *and there exists at least* $j \in \{1, 2, \ldots, q\}$, *such that* $F_2^j < F_1^j$.

If the objective value $\breve{F}(x_1)$ of $x_1$ is dominated by the objective value $\breve{F}(x_2)$ of $x_2$, we also denote the relation by $x_2 \prec x_1$. After a decision $x \in R^n$ is made by the leader, the optimal solution set to the follower's problem is defined as follows:

$$O_x = \{y \in R^m : \nexists y' \in R^m, f(x, y') \prec f(x, y)\}$$

Let $Q = \{(x, y) \in B, y \in Q_x\}$. The set is known as an inducible region in BMPPs.

**Definition 0.2** (*Non-dominated solution set*): *For a BMPP, the Non-dominated solution set* $P^*$ *for the leader's problem is defined as*:

$$P^* = \{(x, y) \in Q, \nexists (x', y') \in Q, F(x', y') \prec F(x, y)\}$$

**Definition 0.3** (*Pareto front*): *For BMPP, the Pareto front* $P\mathcal{F}_u^*$ *for the leader's problem is defined as*:

$$P\mathcal{F}_u^* = \{F(x, y), (x, y) \in P^*\}$$

According to the above definitions, BMPPs can be equivalently written as:

$$\begin{cases} \min_x F(x, y) = (F_1(x, y), F_2(x, y), \ldots, F_q(x, y)) \\ s.t. \quad (x, y) \in Q \end{cases} \quad (3)$$

Assumptions:

(A1) For each $x$ taken in $B_1 \cap B_2$ by the leader, the follower has room to react, that is to say, $O_x \neq \emptyset$.

(A2) $F(x, y)$ is differential in $x$, and $f(x, y)$ and $g(x, y)$ is differentiable and convex in $y$ for $x$ fixed.

Here, $A1$ is a common assumption in BLPPs, which makes BLPPs well-posed. $A2$ is presented to conveniently utilize gradient information.

## Algorithm design

### Transformation of follower problem

BMPPs will incur a high computational cost if the follower's problem is addressed as a general multiobjective optimisation. In the proposed approach, the solution of the follower's problem is categorised into two steps. First, multiobjective problem is converted into several single-objective problems by the weighted sum of the objectives. Subsequently, these converted problems are solved using simplified surrogate models that can efficiently decrease the computational cost; this will be presented in the next Section.

1). The uniform design of weights:

Uniform design and spherical coordinates are used to generate weights uniformly distributed in $\left[0, \frac{\pi}{2}\right]^{p-1}$.

As a classical experimental design method, uniform design has become popular since the 1980's. It was originally developed to obtain designed points that are uniformly distributed over the experimental domain [45, 46]. A uniform design array for $k$ factors with $q$ levels and $H$ combinations is often denoted by $L_H(K^q)$. For example, $L_{16}(4^5)$ involves four factors, and each factor has five levels. Therefore, $1024(= 4^5)$ combinations exist. However, in a uniform design, one must only select 16 combinations from those 1024 possible combinations. $H$ selected combinations can be represented by a uniform matrix $U(n, H) = [U_{i\ell}]_{H \times n}$, where $U_{i\ell}$ is the level of the $\ell^{th}$ factor in the $i^{th}$ combination. In this subsection, we use the concept of uniform design to generate uniformly distributed points in $\left[0, \frac{\pi}{2}\right]^{p-1}$. For a closed and bounded set $\varphi \subset R^{p-1}$, the main purpose of the uniform design is to sample a small group of points from set $\varphi$, such that the sampled points are uniformly scattered in set $\varphi$. Herein, we consider only sample points in the set $\left[0, \frac{\pi}{2}\right]^{p-1}$. A brief description of the uniform design in the set is presented as follows.

For any point $\theta = (\theta_1, \theta_2, \ldots, \theta_{p-1})$ in $\varphi$, a hyper-rectangle between 0 and $\theta$, denoted by $\varphi(\theta)$, can be defined as [40]:

$$\varphi(\theta) = \{(r_1, r_2, \ldots, r_{p-1}) | 0 \leq r_i \leq \theta_i, i = 1, 2, \ldots, p - 1\} \tag{4}$$

For a set of $v$ points on $\varphi$, we assume that $v(\theta)$ points exist in the hyper-rectangle $\varphi(\theta)$. Therefore, the ratio of points in the hyper-rectangle $\varphi(\theta)$ is $\frac{v(\theta)}{v}$, the volume of the hyper-rectangle is $\left(\frac{\pi}{2}\right)^{p-1}$, and the percentage of the hyper-rectangle volume is $\frac{\theta_1 \theta_2 \ldots \theta_{p-1}}{\left(\frac{\pi}{2}\right)^{p-1}}$. Subsequently, the uniform design on $\varphi$ is defined as determining $v$ points on $\varphi$ such that the following discrepancy is minimised.

$$\sup_{\theta \in \varphi} \left| \frac{v(\theta)}{v} - \frac{\theta_1 \theta_2 \ldots \theta_{p-1}}{\left(\frac{\pi}{2}\right)^{p-1}} \right| \tag{5}$$

To determine uniformly scattered points on $\varphi$, we used the good-lattice-point method [47] to generate a set of $v$ uniformly scattered points on $\varphi$, denoted by $c(p - 1, v)$. The procedure is as follows: a $v \times (p - 1)$ uniform array is first generated:

$$U(p - 1, v) = [U_{\ell i}]_{v \times (p-1)} \tag{6}$$

where $U_{\ell i} = (\ell \sigma^{i-1} \bmod v) + 1$, $i = 1, 2, \ldots, p - 1$, $\ell = 1, 2, \ldots, v$. Then, each row of matrix $U(p - 1, v)$ can define a row $\theta^\ell = (\theta_{1\ell}, \theta_{2\ell}, \ldots, \theta_{p-1, \ell})$ of $c(p - 1, v)$ by

$$\theta_{i\ell} = \frac{2U_{\ell i} - 1}{2v} \frac{\pi}{2}, i = 1, 2, \ldots, p - 1, \ell = 1, 2, \ldots, v \tag{7}$$

Hence, we have

$$c(p - 1, v) = \{\theta^\ell | \ell = 1, 2, \ldots, v\} \tag{8}$$

**Table 1. Values of parameter $\sigma$ for different values of $q$ and $M$.**

| q | M | $\sigma$ |
|---|---|---|
| 5 | 2-4 | 2 |
| 7 | 2-6 | 3 |
| 11 | 2-10 | 7 |
| 13 | 2 | 5 |
| | 3 | 4 |
| | 4-12 | 6 |

For example, when $v = 7$ and $p = 5$, it can be seen that $\sigma = 3$ from Table 1, Thus:

$$U(p - 1, v) = U(4, 7) = \begin{pmatrix} 2 & 4 & 3 & 7 \\ 3 & 7 & 5 & 6 \\ 4 & 3 & 7 & 5 \\ 5 & 6 & 2 & 4 \\ 6 & 2 & 4 & 3 \\ 7 & 5 & 6 & 2 \\ 1 & 1 & 1 & 1 \end{pmatrix}_{7 \times 4} \tag{9}$$

$$c(p - 1, v) = \{\frac{\pi}{28}(3, 7, 5, 13), \frac{\pi}{28}(5, 13, 9, 11), \frac{\pi}{28}(7, 5, 13, 9), \frac{\pi}{28}(9, 11, 3, 7),$$
$$\frac{\pi}{28}(11, 3, 7, 5), \frac{\pi}{28}(13, 9, 11, 3), \frac{\pi}{28}(1, 1, 1, 1)\} \tag{10}$$

2). Weight vector
We can obtain $v$ weight vectors as follows:

$$w_\ell = (w_{1\ell}, w_{1\ell}, \ldots, w_{p\ell}), \ell = 1, 2, \ldots, v \tag{11}$$

where if $p = 2$, we have

$$\begin{cases} w_{1\ell} = \cos^2\theta_{p-1\ell} \\ w_{2\ell} = \sin^2\theta_{p-1\ell}, \ \ell = 1, 2, \ldots, v \end{cases} \tag{12}$$

otherwise, if $p > 2$, we also have

$$\begin{cases} w_{1\ell} = \cos^2\theta_{1\ell} \\ w_{2\ell} = \sin^2\theta_{1\ell}\cos^2\theta_{2\ell} \\ w_{3\ell} = \sin^2\theta_{1\ell}\sin^2\theta_{2\ell}\cos^2\theta_{3\ell} \\ w_{4\ell} = \sin^2\theta_{1\ell}\sin^2\theta_{2\ell}\sin^2\theta_{3\ell}\cos^2\theta_{4\ell} \qquad , \ell = 1, 2, \ldots, v \\ \cdots \cdots \cdots \cdots \\ w_{p-1,\ell} = \sin^2\theta_{1\ell}\sin^2\theta_{2\ell}\cdots\sin^2\theta_{p-2\ell}\cos^2\theta_{p-1\ell} \\ w_{p,\ell} = \sin^2\theta_{1\ell}\sin^2\theta_{2\ell}\cdots\sin^2\theta_{p-2\ell}\sin^2\theta_{p-1\ell} \end{cases} \tag{13}$$

3). Transformed problems

We used the weight vectors designed in the above to address the follower's multiple objectives, transforming the follower's problem of the original problem into several single-objective ones. Subsequently, then (2) can be transformed into $v$ BMPPs with a single objective in their follower's problems:

$$\begin{cases} \min_x F(x, y) = (F_1(x, y), F_2(x, y), \ldots, F_q(x, y)) \\ s.t. G_k(x, y) \leq 0, k = 1, 2, \ldots, K \\ \min_y f(x, y) = w_{1\ell} f_1(x, y) + w_{2\ell} f_2(x, y) + \cdots + w_{p\ell} f_p(x, y) \quad , \ell = 1, 2, \cdots, v \\ s.t. g_j(x, y) \leq 0, j = 1, 2, \ldots, J \end{cases} \quad (14)$$

The follower's problems of (14) are:

$$\begin{cases} \min_y f(x, y) = w_{1\ell} f_1(x, y) + w_{2\ell} f_2(x, y) + \cdots + w_{p\ell} f_p(x, y) \\ s.t. G_k(x, y) \leq 0, k = 1, 2, \ldots, K \quad\quad , \ell = 1, 2, \cdots, v \\ \quad g_j(x, y) \leq 0, j = 1, 2, \ldots, J \end{cases} \quad (15)$$

## Surrogate models

In BLPPs/BMPPs, for each the leader's variable values, the follower's problem must be optimised. The procedure result in can a significant amount of computation in solving BLPPs/BMPPs, particularly when the problem is large. According to the optimisation procedure, the optimal solutions to the follower's problem are always determined by the leader's variables. This means that the optimal solution of the follower's problem is a function of the leader's variables. However, the function is often implicit and can not be obtained analytically. In the proposed approach, we used the interpolation function as surrogate models to estimate the optimal solutions to the follower's problems. Therefore, some values $x_i$ of the leader's variables are first used, and then the follower's problems are optimised individually. The optimal solutions are denoted as $y_i$, point pairs $(x_i, y_i)$ are regarded as interpolation sample points and used to generate interpolation functions.

The interpolation function demonstrates better performance in fitting unknown functions [48] and can efficiently decrease the computational times of the follower's problems. In the proposed algorithm, the cubic spline interpolation method is adopted when the interpolation function is one-dimensional, whereas the linear interpolation method is used for other cases.

As an example, on the one-dimensional coordinate plane, for $w$ sample points $x_i$, $i = 1, 2, \ldots, w$, we constructed an interpolation function $y = \psi(x)$ to approximate the optimal solution function of the follower's problem. Therefore, $y = \psi(x^*)$ can be regarded as the approximate solution to the follower's problem when $x$ is fixed at $x^*$. The relationship between the approximate and real solutions is shown in Fig 1.

For interpolation functions, denser sample interpolation points resulted in a better approximation effect. It is noteworthy that for these interpolation points, each follower's variable value must be optimal when the leader's components are fixed. In the proposed algorithm, the interpolation function can be obtained as follows. First, an initial population of $N$ points is randomly generated for the leader's variable space, and these points are denoted by $x_i$, $i = 1, 2, \ldots, N$. Subsequently, for each leader's value among $x_i$, $i = 1, 2, \ldots, N$, the optimal solutions to the follower's problem (15) are denoted by $y_{\ell i}$, $i = 1, 2, \ldots, N$, $\ell = 1, 2, \ldots, v$. Consequently, $v \times N$ point pairs of $(x_i, y_{\ell i})$, $i = 1, \ldots, N$, $\ell = 1, 2, \ldots, v$ can be obtained. These point pairs are used as

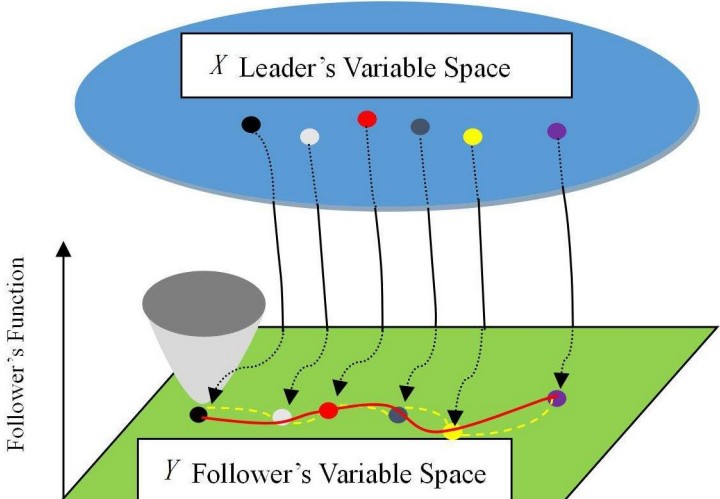

**Fig 1. Interpolation approximation and the real optimal solutions.** In Fig 1, the solid red line represents the real solutions to the follower's problem, whereas the dotted yellow line represents the approximate points provided by interpolation.

interpolation nodes to generate $v$ interpolation functions.

$$\psi_\ell(x) = (\psi_\ell^1(x), \psi_\ell^2(x), \ldots, \psi_\ell^m(x)), \ \ell = 1, 2, \ldots, v \qquad (16)$$

Where

$$\begin{cases} y_\ell^1 \approx \psi_\ell^1(x^1, x^2, \ldots, x^n) \\ y_\ell^2 \approx \psi_\ell^2(x^1, x^2, \ldots, x^n) \\ \ldots \quad \ldots \quad \ldots \\ y_\ell^m \approx \psi_\ell^m(x^1, x^2, \ldots, x^n) \end{cases}, \ \ell = 1, 2, \ldots, v \qquad (17)$$

i.e. each of $y_\ell^j, j = 1, \ldots, m, \ell = 1, \ldots, v$, is a function of $x$ and $y_l(x) = (y_\ell^1, y_\ell^2, \ldots, y_\ell^m), \ell = 1, \ldots, v$. According to the above-mentioned interpolation method, we can obtain the approximate optimal solutions to the follower's problem (15). In the proposed algorithm, we periodically update both the interpolation sample points and to improve interpolation functions.

## Proposed algorithm

When solving BMPPs, Transformation of follower problem guarantees the reduction of the computational complexity of the follower's problems, surrogate models guarantees the saving of the calculation cost of the follower's problems. Combining the methods in above, in this paper, an evolutionary algorithm [49–51]. based on surrogate models, denoted by SMEA, is developed to solve BMPPs. Fig 2 gives the flowchart of SMEA.

The specific procedure is described as follows:

**Step 0** (Transformation of the follower's problems)

The weight vectors uniformly designed in section 3 are used to deal with the follower's multiple objectives, making the follower's problem become $v$ single-objective ones. As a result, we obtain $v$ BMPPs with different follower's objectives.

**Step 1** (Initial population)

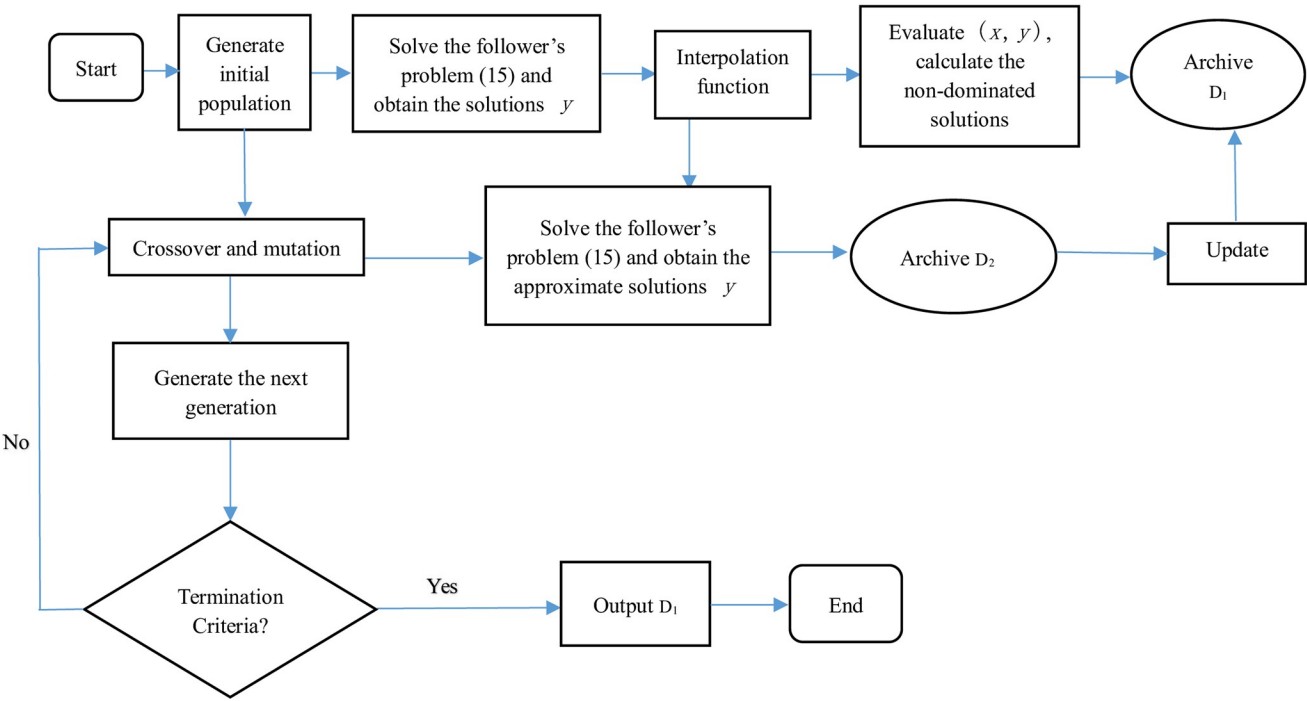

**Fig 2. Basic flowchart of the proposed algorithm.**

Randomly generate $N$ initial points $x_i = a_i + (b_i − a_i) \cdot rand$, $i = 1, \ldots, N$, where $b_i$, and $a_i$ are the upper and lower bounds of the $x_i$, respectively, $rand$ is random in $[0, 1]$. Then the initial population $pop(0)$ with size $N$ is obtained. Set $gen = 0$.

**Step 2** (Fitness assessment)

For each $x_i$, to solve the follower's problems (15) and obtain the optimal solution $y_{\ell i}$, $i = 1, \ldots, N$, $\ell = 1, 2, \ldots, v$. and the value of the leader's objectives are taken as $F_k(x_i, y_{\ell i})$, $i = 1, \ldots, N$, $\ell = 1, 2, \ldots, v$, $k = 1, 2, \ldots, q$. Construct the interpolation functions (surrogate models) just as in Section 4. Use MOEA/D to deal with the leader's problem, a multi-objective optimisation, and take non-dominated solutions in the population $pop(gen)$ into the set $D_1$.

**Step 3** (Crossover)

Note that $F_k(x, y)$, $k = 1, 2, \ldots, q$ are differential in $x$. To obtain a potential descent direction, we take the negative gradient direction of each leader's function $F_k(x_r, y_r)$.

a. Negative gradient vector:

$$-\nabla F_k(x_r, y_r) = -\left(\frac{\partial F_k(x_r, y_r)}{\partial x_1}, \frac{\partial F_k(x_r, y_r)}{\partial x_2}, \cdots, \frac{\partial F_k(x_r, y_r)}{\partial x_n}\right), k = 1, 2, \ldots, q$$

b. Normalize the direction:

$$p_k = -\nabla F_k(x_r, y_r)/ \parallel \nabla F_k(x_r, y_r) \parallel$$
$$= -\left(\frac{\partial F_k(x_r, y_r)}{\partial x_1}, \frac{\partial F_k(x_r, y_r)}{\partial x_2}, \cdots, \frac{\partial F_k(x_r, y_r)}{\partial x_n}\right)/\left(\sum_{j=1}^{n}\left(\frac{\partial F_k(x_r, y_r)}{\partial x_j}\right)^2\right)^{\frac{1}{2}}, k = 1, 2, \ldots, q$$

Set

$$p = \sum_{k=1}^{q} \tau_k p_k$$

Here, $\tau_k \in (0, 1)$, $k = 1, \cdots, q$, are randomly generated and satisfy $\sum_{k=1}^{q} \tau_k = 1$.

c. Crossover operator is designed as follows:

$$x'_r = x_r + s \cdot rand \cdot p$$

where $s$ is positive. In fact, the operator can be extended to non-differential case of the leader's functions. One only needs to replace the gradient function by an approximate gradient:

$$d_{ki} = \frac{F_k(x_1, \cdots, x_i + \Delta d_i, \cdots, x_n) - F_k(x_1, x_2 \cdots, x_i, \cdots, x_n)}{\Delta d_i}, k = 1, 2, \ldots, q$$

**Step 4** (Mutation)

Gaussian mutation is adopted. Suppose that $\bar{p}$ is an individual chosen for mutation, then the offspring $O_{\bar{p}}$ of $\bar{p}$ is generated as follows:

$$O_{\bar{p}} = \bar{p} + \Delta, \Delta \sim N(0, \sigma^2).$$

**Step 5** (Offspring population $pop'(gen)$)

For offspring set $(x_{o1}, x_{o2}, \ldots, x_{o\lambda})$ generated by the crossover and mutation operation, interpolation function is used to obtain the approximated solutions $(y'_{\ell 1}, y'_{\ell 2}, \ldots, y'_{\ell \lambda})$, $\ell = 1, 2, \ldots, v$ to (15). Then an offspring set $pop'(gen)$ with size $v \times \lambda$ is obtained and the values of the leader's objective functions are $F_k(x_{oi}, y_{\ell i})$, $i = 1, 2, \ldots, \lambda$, $\ell = 1, 2, \ldots, v$, $k = 1, 2, \ldots, q$.

**Step 6** (Update interpolation function)

Use MOEA/D to select non-dominated solutions from set $\{(x_{oi}, y_{\ell i}), i = 1, 2, \ldots, \lambda, \ell = 1, 2, \ldots, v\}$, and put these non-dominated solutions into the set $D_2$. For each point in $D_2$, update the optimal solutions $y_{\ell i}$ by solving the follower's problems. And update the interpolation function with the exact solutions obtained above.

**Step 7** (Update Non-dominated solution set $D_1$)

Set $D_1 = D_1 \bigcup D_2$ and $D_2 = \phi$. Remove the dominant solutions in $D_1$. Once the scale of $D_1$ exceeds the pre-determined threshold value, then the crowding distance method can be applied to delete some redundant points just as done in NSGA-II.

**Step 8** (Selection)

Select the best $N$ individuals from set $pop'(gen) \bigcup D_1$ to form the next generation of population $pop(gen + 1)$;

**Step 9** (Termination condition)

If the stopping criterion is satisfied, then stop and output the non-dominated solutions set $D_1$; otherwise, set $gen = gen + 1$, go to Step 3.

## Simulation results

### Test examples

To demonstrate the feasibility and efficiency of the proposed algorithm, we test SMEA on 20 Examples which are taken from literatures [26, 31, 32, 52] and. In [26] and [53], two EAs have been developed. In spite of the fact that the two algorithms are proposed only for dealing with

BMPPs in which the follower's problem is single objective, as a special case, the two approaches can be taken as compared algorithms to demonstrate the performance of the proposed SMEA.

According to the number of objectives, we have divided the test Examples into two categories. One (Examples 1-13) is the bilevel multiobjective case, that is to say, at least one between the leader and the follower involves multiobjective optimisation. The problems of this type are mainly used to test the performance of the weighted sum method and the interpolation approximation. The other (Examples 14-20) only involves bilevel single objective optimization which is utilized to test the performance of surrogate models as well as crossover operators. All 20 examples are presented as follows:

Example 1(F01) [26]:

$$
\begin{cases}
\min\limits_{x} F(x, y) = ((x + 2y_2 + 3)(3y_1 + 2), \ (2x + y_1 + 2)(y_2 + 1)) \\
s.t. \quad 3x + y_1 + 2y_2 \le 5, \quad y_1 + y_2 \le 3 \\
\min\limits_{y} f(x, y) = (y_1 + 1)(x + y_1 + y_2 + 3) \\
\qquad x + 2y_1 + y_2 \le 2, 3y_1 + 2y_2 \le 6 \\
\qquad x \ge 0, y_1 \ge 0, y_2 \ge 0
\end{cases}
$$

Example 2(F02) [26]:

$$
\begin{cases}
\min\limits_{x} F(x, y) = (-2x, \ -x + 5y) \\
\min\limits_{y} f(x, y) = -y \\
s.t. \quad x - 2y \le 4, 2x - y \le 24, 3x + 4y \le 96, x + 7y \le 126 \\
\qquad -4x + 5y \le 65, x + 4y \ge 8, x \ge 0, y \ge 0
\end{cases}
$$

Example 3(F03) [26]:

$$
\begin{cases}
\max\limits_{x} F(x, y) = (2x_1 - 4x_2 + y_1 - y_2, \ -x_1 + 2x_2 - y_1 + 5y_2) \\
\max\limits_{y} f(x, y) = 3y_1 + y_2 \\
s.t. \quad 4x_1 + 3x_2 + 2y_1 + y_2 \le 60, 2x_1 + x_2 + 3y_1 + 4y_2 \le 60 \\
\qquad x_1, x_2, y_1, y_2 \ge 0
\end{cases}
$$

Example 4(F04) [26]:

$$
\begin{cases}
\max\limits_{x} F(x, y) = ((y_1 + y_3)(200 - y_1 - y_3), \ (y_2 + y_4)(160 - y_2 - y_4)) \\
s.t. \quad x_1 + x_2 + x_3 + x_4 \le 40 \\
\qquad 0 \le x_1 \le 10, 0 \le x_2 \le 5, 0 \le x_3 \le 15, 0 \le x_4 \le 20 \\
\min\limits_{y_1, y_2, y_3, y_4} f(x, y) = ((y_1 - 4)^2 + (y_2 - 13)^2, \ (y_3 - 35)^2 + (y_4 - 2)^2) \\
\qquad 0.4y_1 + 0.7y_2 - x_1 \le 0, \ 0.6y_1 + 0.3y_2 - x_2 \le 0 \\
\qquad 0.4y_3 + 0.7y_4 - x_3 \le 0, \ 0.6y_3 + 0.3y_4 - x_4 \le 0 \\
\qquad 0 \le y_1 \le 20, 0 \le y_2 \le 20, 0 \le y_3 \le 40, 0 \le y_4 \le 40
\end{cases}
$$

Example 5(F05) [26]:

$$
\begin{cases}
\min_{x} F(x,y) = (-x - y, \ x^2 + (y - 10)^2) \\
s.t. \quad 0 \le x \le 15 \\
\min_{y} f(x,y) = y(x - 30) \\
\qquad y - x \le 0, 0 \le y \le 15
\end{cases}
$$

Example 6(F06) [26]:

$$
\begin{cases}
\min_{x} F(x,y) = (8x + 4y_1{}^2 - 2, \ 4x - 8y_2 + 1) \\
s.t. \quad 1 \le x \le 2 \\
\min_{y_1,y_2} f(x,y) = (2y_1{}^3 - x + 7, \ x^2 + 2x + y_2{}^2 - 5) \\
\qquad x - y_1 - 1 \le 0, \ x - y_2 \le 0
\end{cases}
$$

Example 7(F07) [26]:

$$
\begin{cases}
\min_{x} F(x,y) = (y_1 + y_2{}^2 + x + \sin^2(y_1 + x), \ \cos(y_2)(0.1 + x)exp(\dfrac{-y_1}{0.1 + y_2})) \\
s.t. \quad 0 \le x \le 10 \\
\min_{y_1,y_2} f(x,y) = \dfrac{(y_1 - 2)^2 + (y_2 - 1)^2}{4} + \dfrac{y_2 x + (5 - x)^2}{16} + \sin\dfrac{y_2}{10} + \\
\qquad \dfrac{y_1{}^2 + (y_2 - 6)^2 - 2y_1 x - (5 - x)^2}{80} \\
\qquad y_1{}^2 - y_2 \le 0, \ 5y_1{}^2 + y_2 \le 10, \ y_2 + \dfrac{x}{6} \le 5, \ y_1 \ge 0
\end{cases}
$$

Example 8(F08) [26]:

$$
\begin{cases}
\min_{x} F(x,y) = ((y_1 - 1)^2 + \sum_{i=1}^{13} y_{i+1}^2 + x^2, (y_1 - 1)^2 + \sum_{i=1}^{13} y_{i+1}^2 + (x - 1)^2) \\
\min_{y} f(x,y) = (y_1 - x)^2 + \sum_{i=1}^{13} y_{i+1}^2 \\
s.t. \quad -1 \le x, y_i \le 2, \ i = 1, 2, \dots, 14
\end{cases}
$$

Example 9(F09) [26]:

$$
\begin{cases}
\min_{x} F(x,y) = ((1 - x_1)(1 + x_2{}^2 + x_3{}^2)y, \ x_1(1 + x_2{}^2 + x_3{}^2)y) \\
\min_{y} f(x,y) = (1 - x_1)(1 + x_4{}^2 + x_5{}^2)y \\
s.t. \quad (1 - x_1)y + \dfrac{1}{2}x_1 y - 1 \ge 0, \\
\qquad -1 \le x_1 \le 1, \ 1 \le y \le 2, -5 \le x_i \le 5, \ i = 2, 3, 4, 5
\end{cases}
$$

Example 10(F10) [26]:

$$
\begin{cases}
\min_x F(x,y) = \left( \sum_{i=1}^{13} exp(\dfrac{-x_i}{1+\mid y_i \mid}) + \sum_{i=1}^{13} sin(\dfrac{x_i}{1+\mid y_i \mid}), \right. \\
\qquad\qquad \sum_{i=1}^{13} exp(\dfrac{-y_i}{1+\mid x_i \mid}) + \sum_{i=1}^{13} sin(\dfrac{y_i}{1+\mid x_i \mid}) \left. \right) \\
s.t. \qquad -1 \le x_i \le 1, \ i = 1,2,\ldots,13 \\
\min_y f(x,y) = \sum_{i=1}^{13} cos(\mid x_i \mid y_i) + \sum_{i=1}^{13} sin(x_i - y_i) \\
\qquad x_i + y_i \le 1, \ i = 1,2,\ldots,13, -1 \le y_i \le 1, \ i = 1,2,\ldots,13
\end{cases}
$$

Example 11(F11) [32]:

$$
\begin{cases}
\min_x F(x,y) = (y_1 - x, \ y_2) \\
\min_y f(x,y) = (y_1, \ y_2) \\
s.t. \quad x^2 - y_1^2 - y_2^2 \ge 0, 1 + y_1 + y_2 \ge 0 \\
\qquad -1 \le y_1, y_2 \le 1, 0 \le x \le 1
\end{cases}
$$

Example 12(F12) [32]:

$$
\begin{cases}
\min_x F(x,y) = \left( (y_1-1)^2 + \sum_{i=1}^{13} y_{i+1}^2 + x^2, \ (y_1-1)^2 + \sum_{i=1}^{13} y_{i+1}^2 + (x-1)^2 \right) \\
\min_y f(x,y) = (y_1^2 + \sum_{i=1}^{13} y_{i+1}^2, \ (y_1-x)^2 + \sum_{i=1}^{13} y_{i+1}^2) \\
s.t. \quad -1 \le y_1, y_2, \ldots, y_{14}, x \le 2
\end{cases}
$$

Example 13(F13) [31]:

$$
\begin{cases}
\min_x F(x,y) = ((1-y_1)(1+\sum_{j=2}^{5} y_j^2 )x), \ y_1(1+\sum_{j=2}^{5} y_j^2 )x) \\
\min_y f(x,y) = ((1-y_1)(1+\sum_{j=6}^{9} y_j^2 )x, \ y_1(1+\sum_{j=6}^{9} y_j^2 )x) \\
s.t. \quad (1-y_1)x + \dfrac{1}{2}y_1 x - 1 \ge 0 \\
\qquad -1 \le y_1 \le 1, 1 \le x \le 2, -9 \le y_j \le 9, j = 2,3,\ldots,9
\end{cases}
$$

Example 14(F14) [52]:

$$
\begin{cases}
\min_x F(x,y) = -15x_1 + 5x_2 - 5x_3 + 21x_4 - 25x_5 + 31x_6 - 19x_7 + 32x_8 + 13x_9 - 13x_{10} \\
\qquad\qquad - 32y_1 - 30y_2 - 15y_3 + 12y_4 - 31y_5 - 3y_6 + 15y_7 - 44y_8 - 38y_9 - 5y_{10} \\
s.t. \quad 8x_1 + 6x_2 + 10x_3 + 10x_4 + 4x_5 + 7x_6 + 7x_7 + 7x_8 + 3x_9 + 7x_{10}+ \\
\qquad 4y_1 + 2y_2 + 7y_3 + 7y_4 + 10y_5 + 8y_6 + 9y_7 + y_8 + 8y_9 + 2y_{10} \le 65 \\
\qquad 9x_1 + x_2 + 10x_3 + 5x_4 + 10x_5 + 8x_7 + x_8 + 3x_{10}+ \\
\qquad 4y_1 + 5y_2 + 8y_3 + y_4 + 3y_5 + 2y_6 + 10y_7 + 2y_8 + 2y_9 + 2y_{10} \le 113 \\
\qquad x_1 + 3x_2 + x_3 + 8x_4 + 8x_5 + 9x_6 + 8x_7 + 7x_8 + x_9 + 10x_{10}+ \\
\qquad 8y_1 + 4y_2 + 3y_3 + y_4 + 6y_5 + 5y_6 + 6y_7 + 9y_8 + 10y_9 + 6y_{10} \le 89 \\
\qquad 10x_1 + 6x_2 + 10x_3 + x_4 + 10x_5 + 10x_6 + 4x_7 + 9x_9+ \\
\qquad 8y_1 + 7y_2 + 7y_3 + 5y_4 + 2y_5 + 7y_6 + y_7 + 2y_8 + 3y_9 + 5y_{10} \le 85 \\
\qquad x_i \in Z, i = 1,2,3,4,5,6,7,8,9,10 \\
\min_y f(x,y) = -32y_1 - 30y_2 - 15y_3 + 12y_4 - 31y_5 - 3y_6 + 15y_7 - 44y_8 - 38y_9 - 5y_{10} \\[4pt]
\qquad 10x_1 + 4x_2 + 5x_3 + 6x_4 + x_5 + x_6 + 7x_7 + 2x_8 + 5x_9 + x_{10}+ \\
\qquad 8y_1 + 2y_2 + 2y_3 + 9y_4 + 9y_5 + 4y_6 + 2y_7 + 9y_8 + 3y_9 + 8y_{10} \le 19 \\
\qquad 3x_1 + 6x_2 + 8x_3 + 5x_4 + 8x_5 + 6x_6 + 8x_7 + 10x_8 + 10x_9 + 10x_{10}+ \\
\qquad 9y_1 + 8y_2 + 2y_3 + 6y_4 + 6y_5 + 2y_7 + 10y_8 + 9y_9 + 4y_{10} \le 23 \\
\qquad 8x_1 + 10x_3 + 3x_5 + 2x_6 + 4x_7 + x_8+ \\
\qquad 4y_2 + y_3 + 6y_4 + 3y_5 + 2y_6 + 4y_7 + 5y_8 + 4y_9 + 2y_{10} \le 105 \\
\qquad 8x_1 + x_3 + 3x_4 + 5x_5 + 7x_6 + 9x_8 + 4x_9 + 8x_{10}+ \\
\qquad 4y_1 + 10y_2 + y_3 + y_4 + 5y_5 + y_6 + 5y_8 + y_9 + 4y_{10} \le 106 \\
\qquad y_i \ge 0, i = 1,2,3,4,5,6,7,8,9,10 \\
\qquad y_j \in Z, j \in J, J = 1,5,9,10
\end{cases}
$$

Example 15(F15) [32]:

$$
\begin{cases}
\min_x F(x,y) = -(200 - x_1 - x_2)(x_1 + x_3) - (160 - x_2 - x_4)(x_2 + x_4) \\
s.t. \quad x_1 + x_2 + x_3 + x_4 \le 40, \\
\qquad 0 \le x_1 \le 10, 0 \le x_2 \le 5, 0 \le x_3 \le 15, 0 \le x_4 \le 20, \\
\min_y f(x,y) = (y_1 - 4)^2 + (y_2 - 13)^2 + (y_3 - 35)^2 + (y_4 - 2)^2 \\
\qquad 0.4y_1 + 0.7y_2 \le x_1, 0.6y_1 + 0.3y_2 \le x_2, \\
\qquad 0.4y_3 + 0.7y_4 \le x_3, 0.6y_3 + 0.3y_4 \le x_4, \\
\qquad 0 \le y_1 \le 20, 0 \le y_2 \le 20, 0 \le y_3 \le 40, 0 \le y_4 \le 40.
\end{cases}
$$

Example 16(F16) [53]:

$$\begin{cases} \min_{x} F(x,y) = -(x_1)^2 - 3(x_2)^2 - 4y_1 + y_2^2 \\ \min_{y} f(x,y) = 2(x_1)^2 + y_1^2 - 5y_2 \\ s.t. \quad (x_1)^2 - 2x_1 + (x_2)^2 - 2y_1 + y_2 \geq -3 \\ \qquad x_2 + 3y_1 - 4y_2 \geq 4, (x_1)^2 + 2x_2 \leq 4, \\ \qquad x_i \geq 0, y_i \geq 0, i = 1, 2 \end{cases}$$

Example 17(F17) [53]:

$$\begin{cases} \min_{x} F(x,y) = -8x_1 - 4x_2 + 4y_1 - 40y_2 - 4y_3 \\ \min_{y} f(x,y) = x_1 + 2x_2 + y_1 + y_2 + 2y_3 \\ s.t. \quad y_2 + y_3 - y_1 \leq 1, 2x_1 - y_1 + 2y_2 - 0.5y_3 \leq 1 \\ \qquad 2x_2 + 2y_1 - y_2 - 0.5y_3 \leq 1 \\ \qquad y_i \geq 0, i = 1, 2, 3, \ \ x_i \geq 0, i = 1, 2 \end{cases}$$

Example 18(F18) [53]:

$$\begin{cases} \min_{x} F(x,y) = (x_1 - 1)^2 + 2y_1 - 2x_1 \\ \min_{y} f(x,y) = (2y_1 - 4)^2 + (2y_2 - 1)^2 + x_1 y_1 \\ s.t. \quad 4x_1 + 5y_1 + 4y_2 \leq 12, 4y_2 - 4x_1 - 5y_1 \leq -4 \\ \qquad 4x_1 - 4y_1 + 5y_2 \leq 4, 4y_1 - 4x_1 + 5y_2 \leq 4 \\ \qquad y_i \geq 0, i = 1, 2, x_1 \geq 0 \end{cases}$$

Example 19(F19) [53]:

$$\begin{cases} \min_{x} F(x,y) = \sum_{i=1}^{10}(|x_i - 1| + |y_i|) \\ \min_{y} f(x,y) = e^{(1+\frac{1}{4000}\sum_{i=1}^{10}(y_i x_i)^2 - \prod_{i=1}^{10} cos(\frac{y_i x_i}{\sqrt{i}}))} \\ s.t. \quad -\pi \leq y_i \leq \pi, i = 1, 2, \ldots\ldots, 10 \end{cases}$$

Example 20(F20) [53]:

$$\begin{cases} \min_{x} F(x,y) = |2x_1 + 2x_2 - 3y_1 - 3y_2 - 60| \\ \min_{y} f(x,y) = (y_1 - x_1 + 20)^2 + (y_2 - x_2 + 20)^2 \\ s.t. \quad 2y_1 - x_1 + 10 \leq 0, 2y_2 - x_2 + 10 \leq 0, \\ \qquad x_1 + x_2 + y_1 - 2y_2 \leq 40 \\ \qquad -10 \leq y_i \leq 20, 0 \leq x_i \leq 50, i = 1, 2 \end{cases}$$

## Parameter setting

To compare the experimental results, the parameter settings in this study are consistent with in [26].

- Population size: 15;

- Weight vector: $v = 10$;

- Crossover rate: $pc = 0.6$;

- Gaussian mutation probability: $pm = 0.05$;

- Maximum number of generations: $maxgen = 300$;

- Step size in the crossover operator: $s = 8$.

## Performance measure

To test the effectiveness and practicability of the SMEA algorithm, we tested the non-dominated solutions, HV, IGD, $C − metric$, CPU time and optimal solution according to the characteristics of the problems. Definitions of several metrics are as follows:

 1). HV [54]

Select a reference point in the objective space $e = (e_1, e_2, \cdots, e_s)^T$, denote A = SMEA, NSGA-II, Weighted sum approach or Tchebycheff approach. Compute

$$HV(A, e) = volume(\bigcup_{f \in A} [f_1, e_1] \times \cdots \times [f_s, e_s])$$

For reference points in the objective space, the large HV means better quality.

 2). IGD [55]

Let $G$ be a uniformly distributed subset selected to form the true Pareto Front and $G'$ is the approximated set that is obtained by a multiobjective optimisation algorithm. The $IGD$ value of $G$ to $G'$, i.e., $IGD(G, G')$ is defined as

$$IGD(G, G') = \frac{\sum_{i=1}^{|G|} d(G_i, G')}{|G|}$$

where $|G|$ returns the number of solutions in the set G and $d(G_i, G')$ computes the minnmum Euclidean distance from $G_i$ to the solutions of $G'$ in objective space. Generally, a lower value of $IGD(G, G')$is preferred as it indicates that $G'$ is distributed more uniformly and closer to the true Pareto front.

 3). $C − metric$ [54]

We give the following notations:

Set $SM$: non-dominated solution set of SMEA;

Set $WS$: non-dominated solution set of Weighted sum approach;

Set $TE$: non-dominated solution set of Tchebycheff approach;

Set $NS$: non-dominated solution set of NSGA-II. Let:

$$C(A', B) = \frac{|\ u \in B\ |\ \exists v \in A' : v\ \text{dominates}\ u\ |}{|\ B\ |}$$

$C(A', B)$ is defined as the percentage of the solutions in B that are dominated by at least one solution in $A'$. $C(A', B)$ is not necessarily equal to $1 − C(B, A')$. $C(A', B) = 1$ implies that no solution in $B$ is dominated by a solution in $A'$.

## Simulation results and comparisons

We execute an algorithm on a computer Intel(R) Core(TM)i5-8250U CPU@ 160 GHz 1.80 GHz using Matlab software. The algorithm was independently run ten times on every test

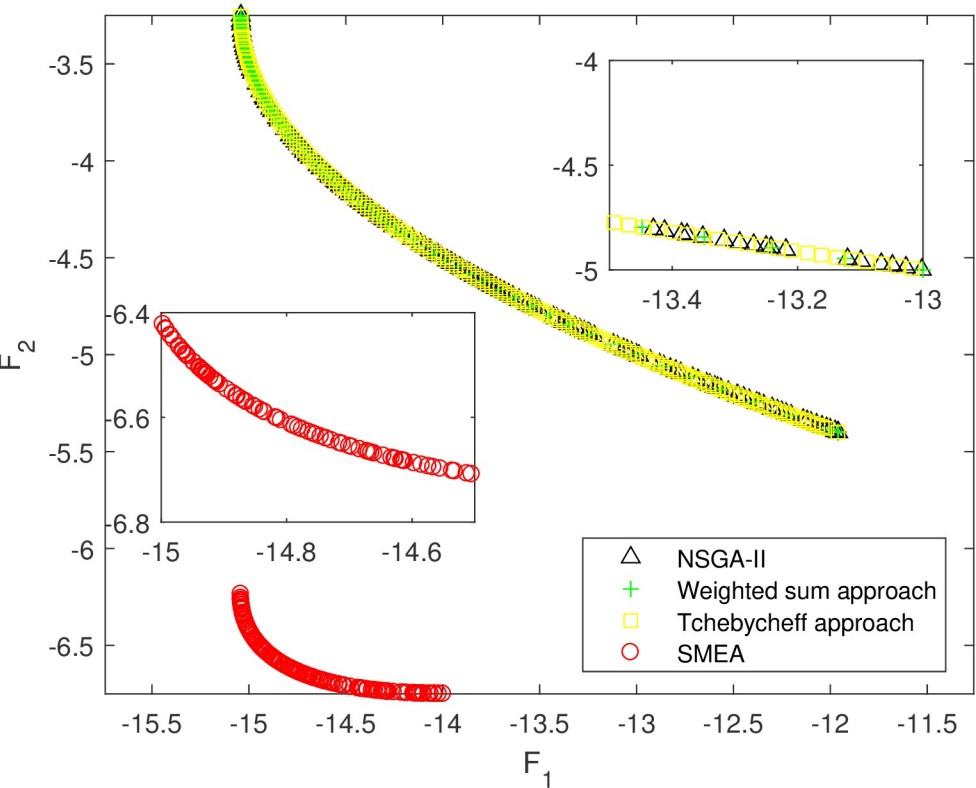

**Fig 3. Non-dominated solutions on example 1.** Comparison of the non-dominated solutions obtained by SMEA, NSGA-II, Weighted sum approach and Tchebycheff approach on example 1.

instance. For the first 10 Examples, SMEA is compared with the weighted sum method, the Tchebycheff approach and NSGA-II. For each algorithm, we randomly take one among all 10 computational results of a single compared algorithm and show them in Figs 3–12. The analytical(theoretical) solution set of Examples 11 to 13 is provided, as a result, we can compare the computational results of SMEA with these known analytical solutions as shown in Figs 13–15.

In Figs 3–12 (example 1–10), for Example 1 (shown in Fig 3), the non-dominated solutions set obtained by SMEA is superior to that obtained by the three compared methods. Example 2 is a maximization problem, from Fig 4 we can see that the non-dominated solutions set obtained by the SMEA is better than those obtained by the three compared methods. In addition, for Example 3, 4, 5, 7 and 9, the non-dominated solutions set obtained by the SMEA is similar to that by the Tchebycheff approach, but better than those by the other two methods. For Example 10, the results presented by the SMEA is almost the same as those obtained by both the Tchebycheff approach and the weighted sum approach, but better than that by the NSGA-II. For Examples 6 and 8, SMEA can also find approximately the non-dominated solutions provided by literature. Figs 13–15 show that the non-dominated solutions sets obtained by SMEA are almost in line with the analytical solutions sets.

Table 2 represents the average value of HV obtained by SMEA, Weighted sum approach, Tchebycheff approach and NSGA-II running 10 times independently on Examples 1–10. While Table 3 represents the average value of HV and IGD obtained by SMEA and Analytical points running 10 times independently on Examples 11–13. The symbols "+", "−" and "≈" mean the computational result is better than, worse than and almost equal to that obtained by

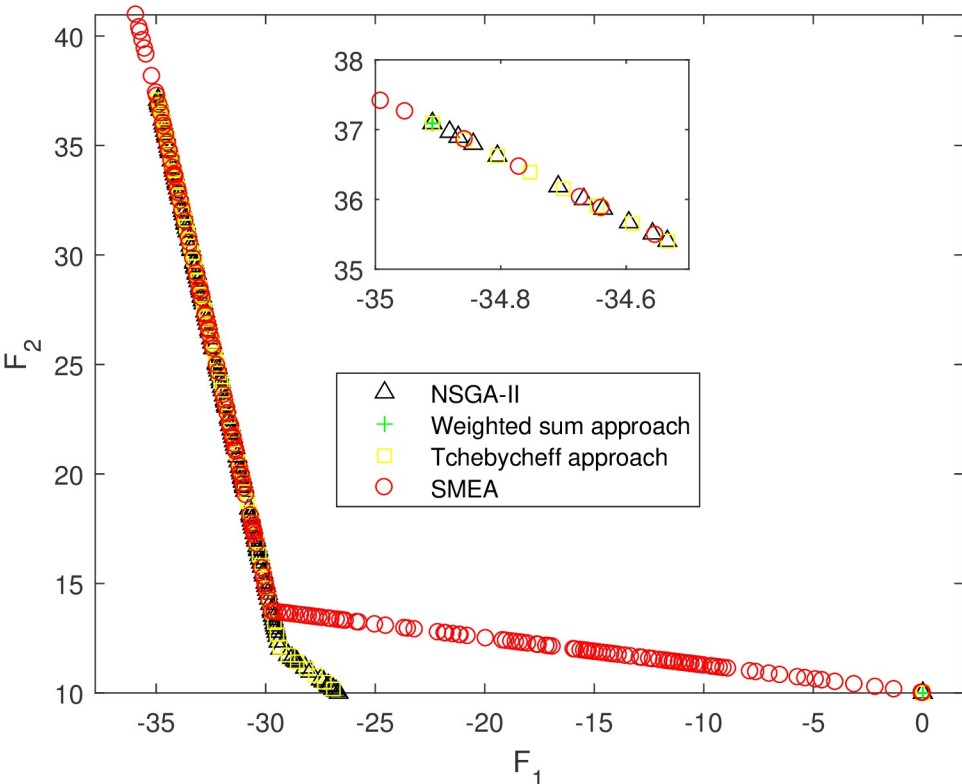

**Fig 4. Non-dominated solutions on example 2.** Comparison of the non-dominated solutions obtained by SMEA, NSGA-II, Weighted sum approach and Tchebycheff approach on example 2.

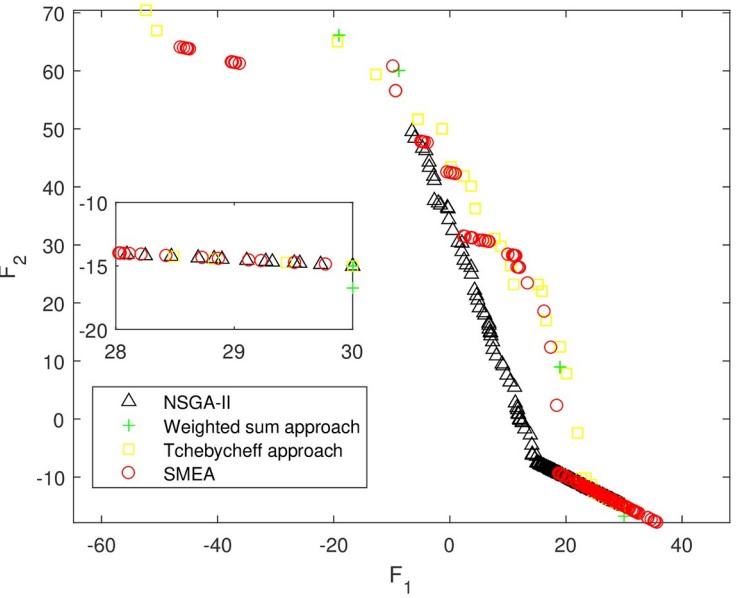

**Fig 5. Non-dominated solutions on example 3.** Comparison of the non-dominated solutions obtained by SMEA, NSGA-II, Weighted sum approach and Tchebycheff approach on example 3.

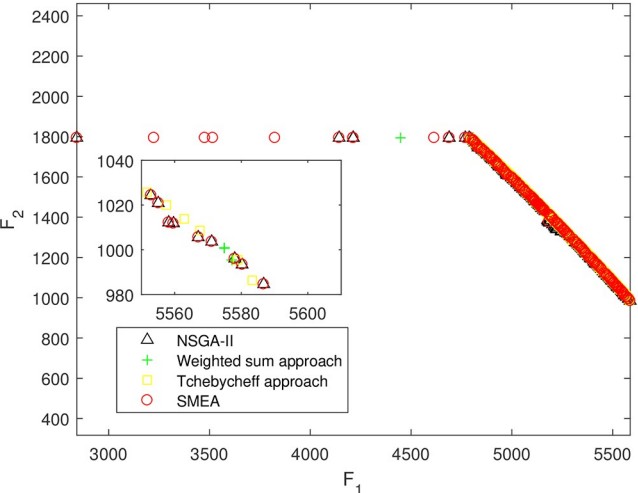

**Fig 6. Non-dominated solutions on example 4.** Comparison of the non-dominated solutions obtained by SMEA, NSGA-II, Weighted sum approach and Tchebycheff approach on example 4.

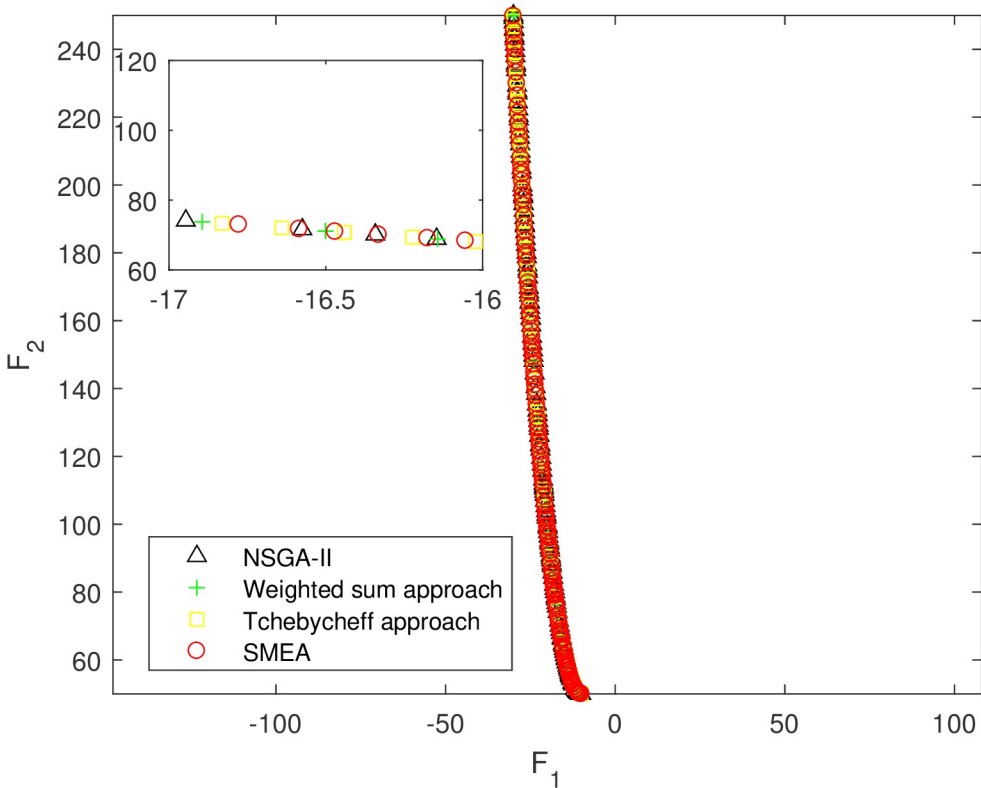

**Fig 7. Non-dominated solutions on example 5.** Comparison of the non-dominated solutions obtained by SMEA, NSGA-II, Weighted sum approach and Tchebycheff approach on example 5.

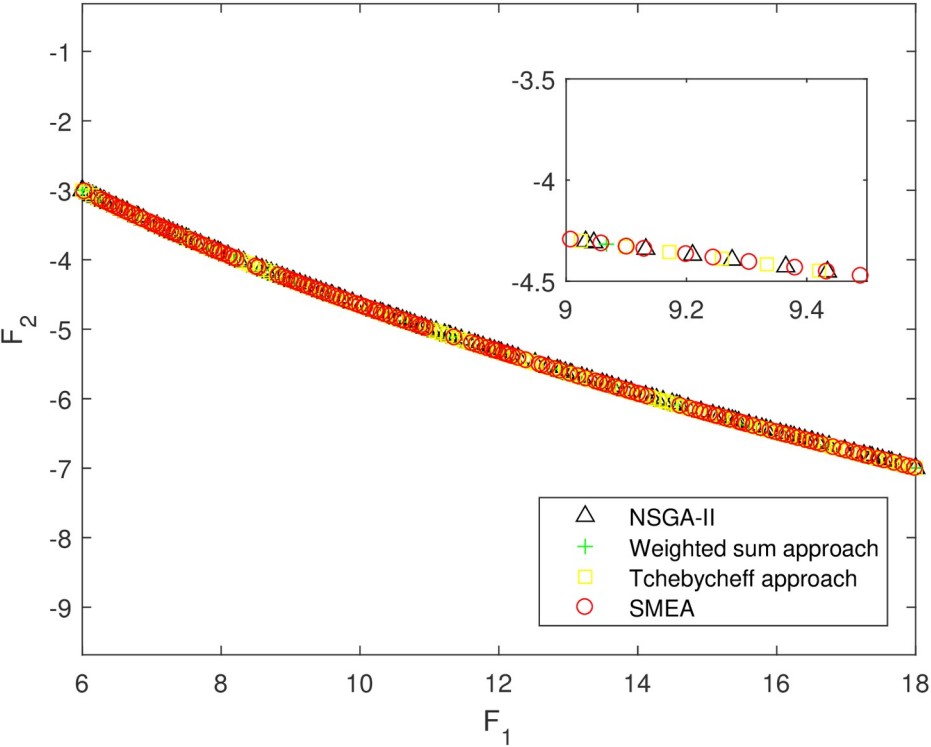

**Fig 8. Non-dominated solutions on example 6.** Comparison of the non-dominated solutions obtained by SMEA, NSGA-II, Weighted sum approach and Tchebycheff approach on example 6.

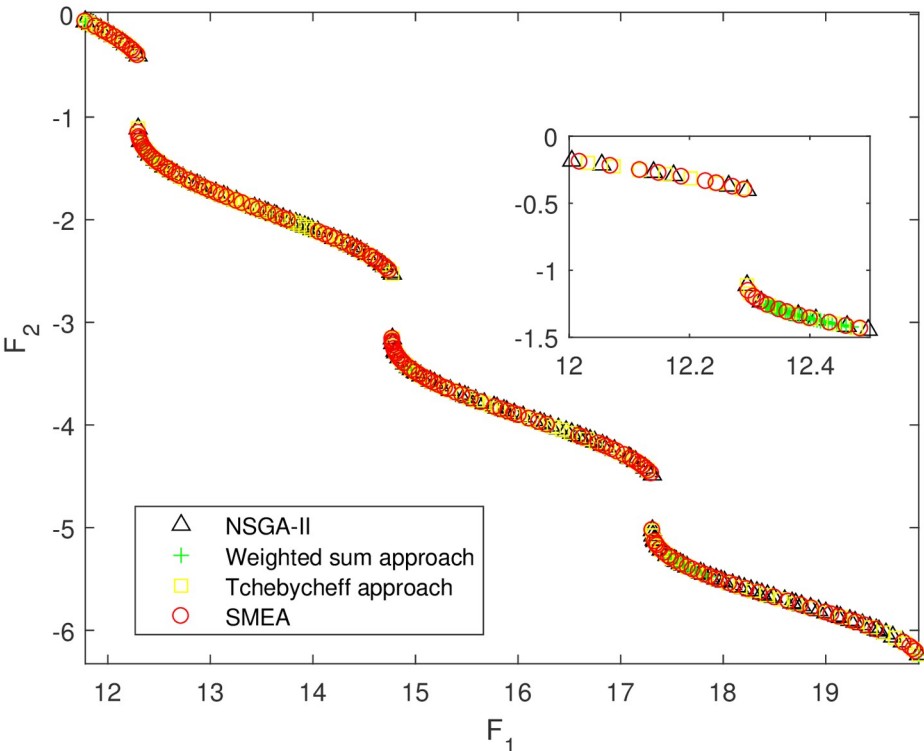

**Fig 9. Non-dominated solutions on example 7.** Comparison of the non-dominated solutions obtained by SMEA, NSGA-II, Weighted sum approach and Tchebycheff approach on example 7.

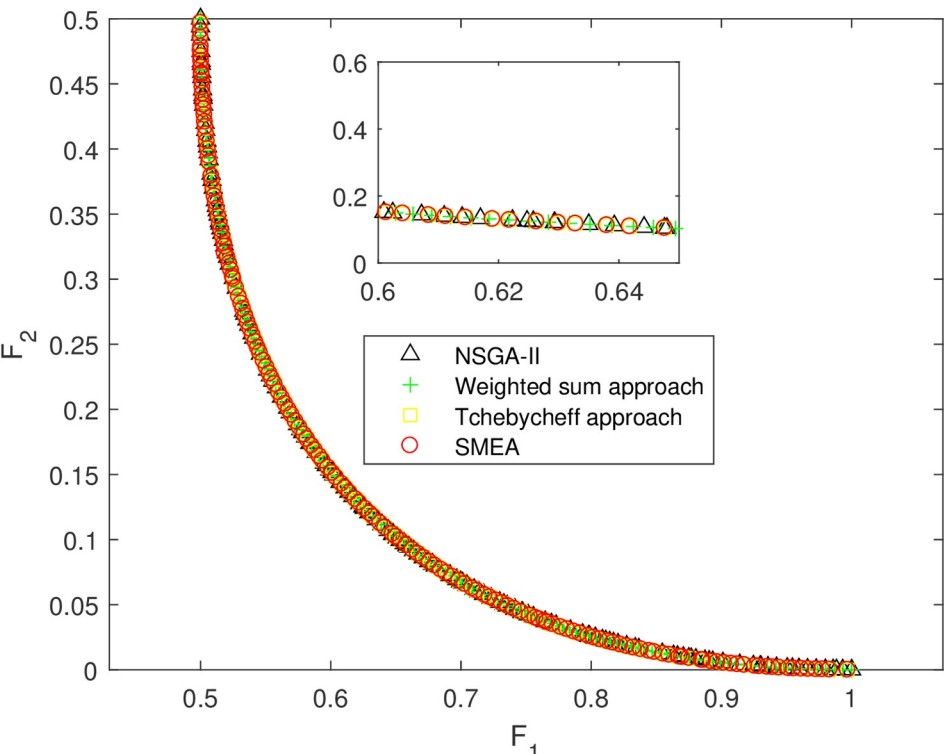

**Fig 10. Non-dominated solutions on example 8.** Comparison of the non-dominated solutions obtained by SMEA, NSGA-II, Weighted sum approach and Tchebycheff approach on example 8.

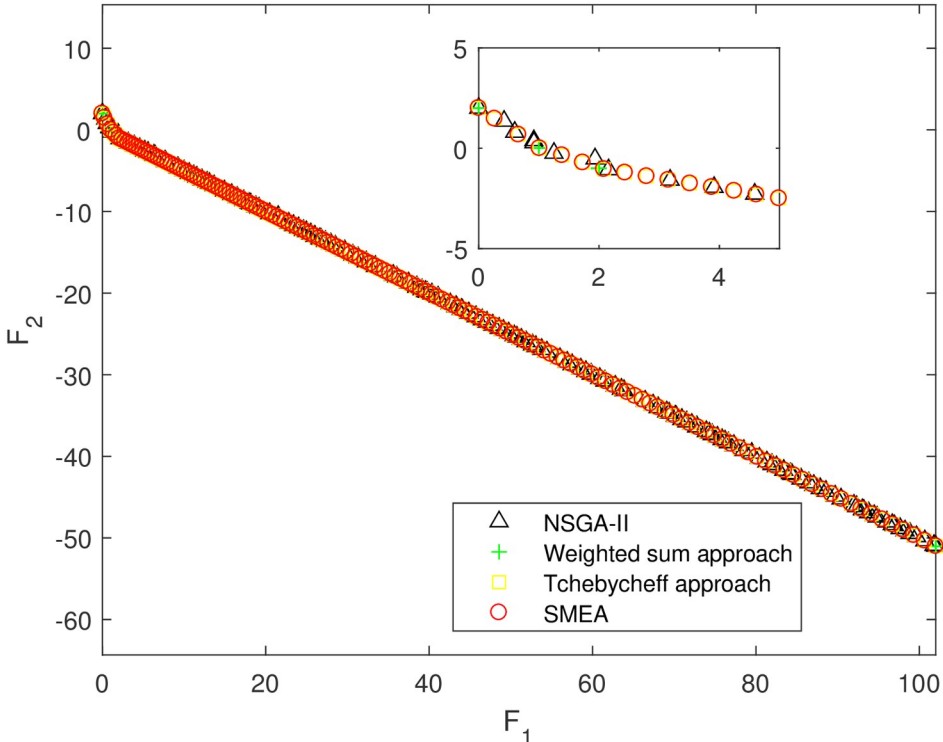

**Fig 11. Non-dominated solutions on example 9.** Comparison of the non-dominated solutions obtained by SMEA, NSGA-II, Weighted sum approach and Tchebycheff approach on example 9.

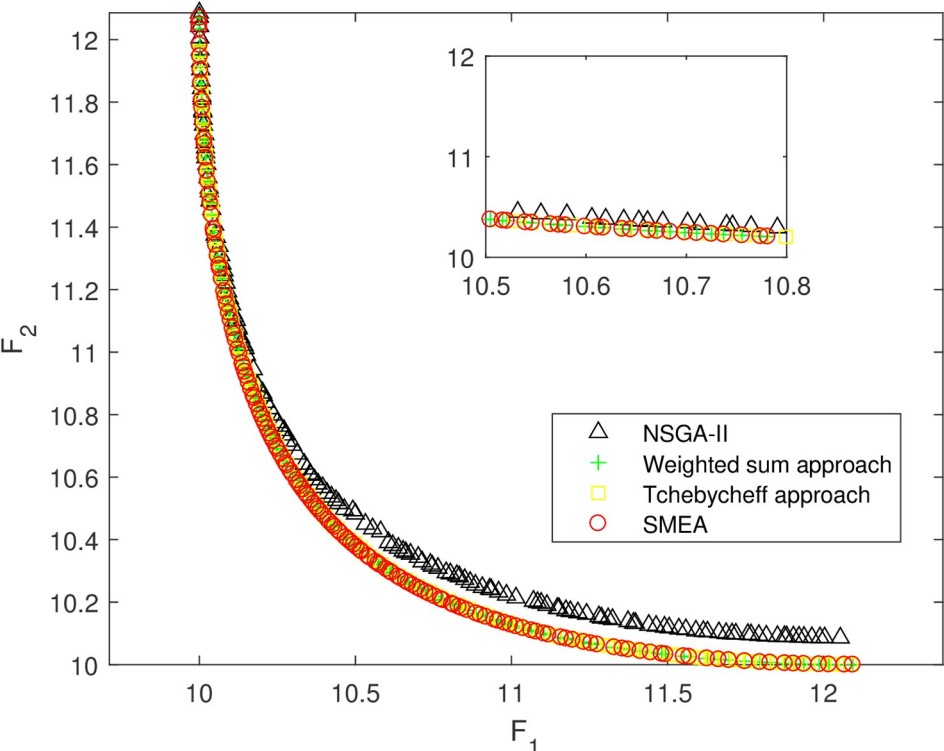

**Fig 12. Non-dominated solutions on example 10.** Comparison of the non-dominated solutions obtained by SMEA, NSGA-II, Weighted sum approach and Tchebycheff approach on example 10.

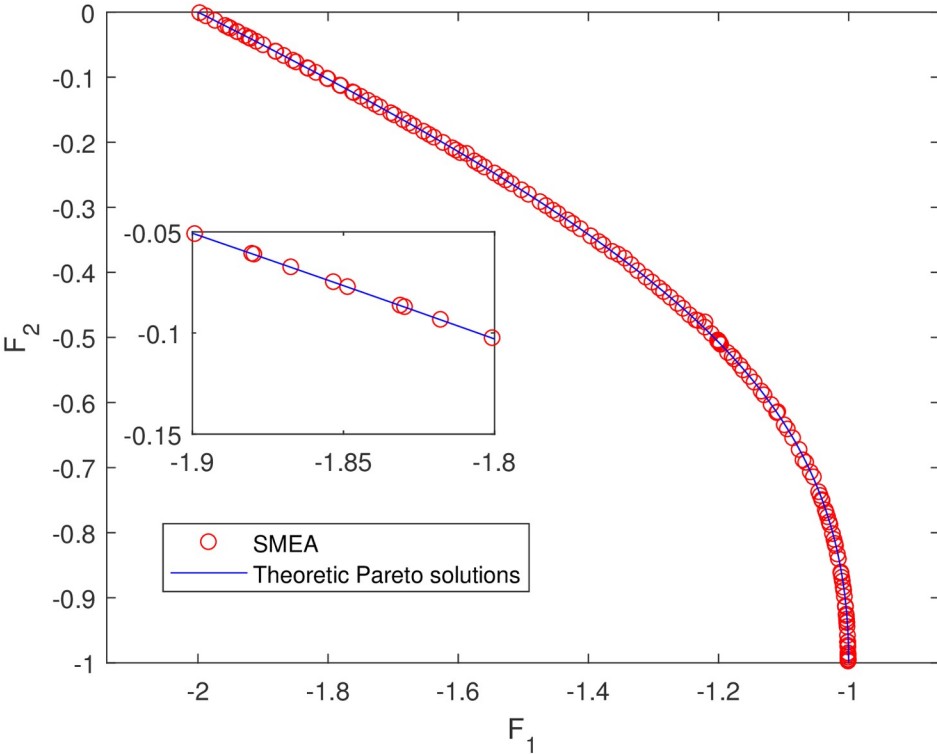

**Fig 13. Non-dominated solutions on example 11.** Analytical solutions and non-dominated solutions obtained by SMEA on example 11.

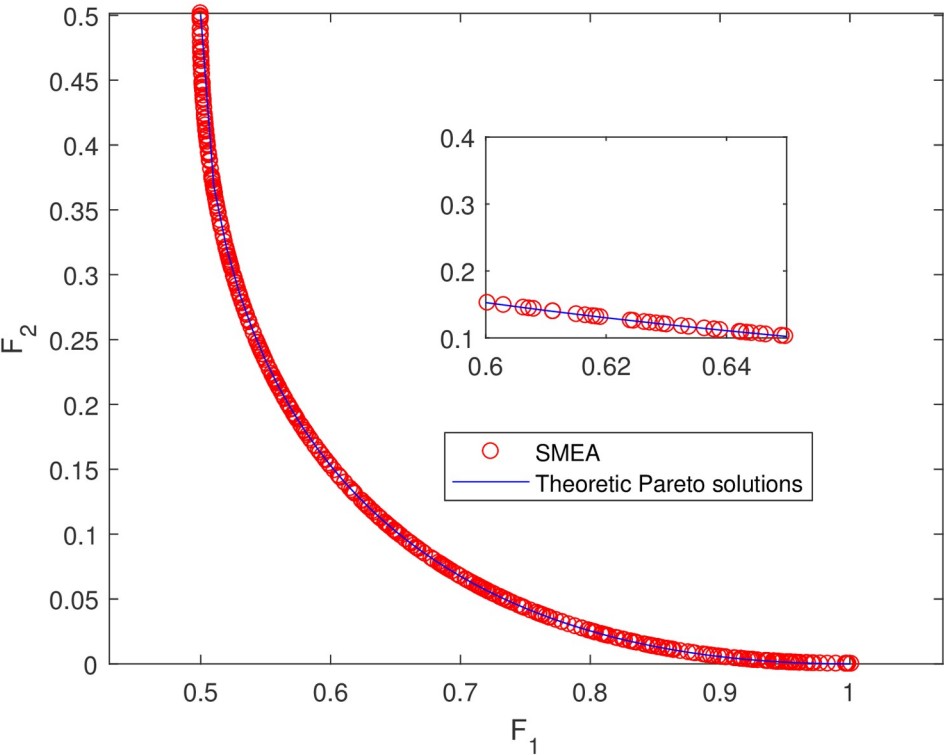

**Fig 14. Non-dominated solutions on example 12.** Analytical solutions and non-dominated solutions obtained by SMEA on example 12.

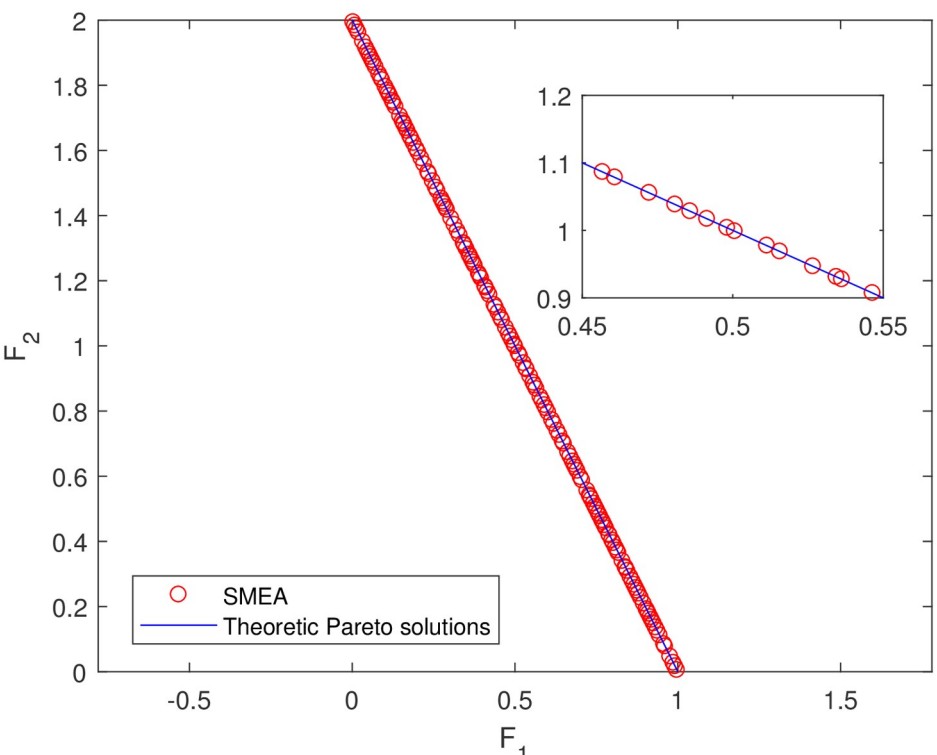

**Fig 15. Non-dominated solutions on example 13.** Analytical solutions and non-dominated solutions obtained by SMEA on example 13.

**Table 2. HV obtain by SMEA and the approaches in the literature.**

| Test problem | SMEA(±std) | Weighted sum approach | Tchebycheff approach | NSGA-II |
|---|---|---|---|---|
| F01 | $3.8841E + 00(\pm 3.1298E - 03)$ | 3.7641E+00 | 3.5249E+00 | 2.4364E+01 |
| F02 | **$1.6698E + 03(\pm 5.7438E - 02)$** | 0.0000E+00 | 8.2293E+01 | 1.2895E+03 |
| F03 | **$6.0071E + 03(\pm 2.2216E - 03)$** | 1.5274E+03 | 4.8412E+03 | 1.4086E+02 |
| F04 | **$7.0414E + 07(\pm 6.0845E - 04)$** | 5.5236E+05 | 3.2912E+05 | 4.6756E+02 |
| F05 | **$2.8771E + 03(\pm 3.5607E - 03)$** | 2.5212E+03 | 2.6888E+03 | 2.7711E+03 |
| F06 | $2.0669E + 02(\pm 1.2233E - 04)$ | 2.5281E+01 | 2.6506E+01 | 3.0948E+02 |
| F07 | **$1.8901E + 03(\pm 4.7892E - 03)$** | 2.5252E+01 | 2.8871E+01 | 1.5958E+03 |
| F08 | $3.2278E + 00(\pm 8.6756E - 03)$ | 2.0780E-01 | 2.0460E-01 | 5.6428E+00 |
| F09 | **$3.2245E + 03(\pm 4.5582E - 03)$** | 3.0200E+02 | 2.7828E+03 | 2.5540E+01 |
| F10 | $3.8894E + 00(\pm 2.5514E - 02)$ | 3.7641E+00 | 3.5249E+00 | 2.4363E+01 |
| $+/-/\approx$ | — | 10/0/0 | 10/0/0 | 6/4/0 |

**Table 3. HV, IGD obtain by SMEA and HV obtain by analytical points.**

| Test problem | SMEA HV(±std) | Analytical points HV | IGD |
|---|---|---|---|
| F11 | $3.1070E - 01(\pm 1.3322E - 03)$ | 3.1160E-01 | 3.4000E-03 |
| F12 | $1.9964E - 01(\pm 4.7843E - 03)$ | 2.0840E-01 | 8.2817E-04 |
| F13 | $9.8030E - 01(\pm 2.53366E - 03)$ | 1.0000E+00 | 4.6000E-03 |
| $+/-/\approx$ | — | 2/0/1 | — |

our algorithm, respectively. Calculated the standard deviation(std)of HV value in 10 times. Best results are highlighted bold color in Table 2, we can see that the HV obtained by SMEA is much larger than those obtained other three algorithms on test problems 2–5, 7, 9 in Table 2. From Table 2, we can see in SMEA 10 values of HV better than the Weighted sum approach, 10 values of HV better than the Tchebycheff approach and 6 values of HV better than NSGA-II. Which illustrates that the diversity of solutions obtained by SMEA is better than obtained by other algorithms. Meanwhile, it is obvious that the IGD obtained by SMEA is small on Examples 11–13, which indicates for these test problems, the coverage of solutions obtained by SMEA quite well to the analytical solutions.

In terms of the results of the multiple problem Wilcoxon's test in Table 4, all the $R^+$ values are bigger than the $R^-$ ones, which implies that the performance of our algorithm is superior to that of other competitors. Moreover, the significant difference at $\alpha = 0.05$ can be found in four cases, i.e., SMEA versus Tchebycheff approach, SMEA versus Weighted sum approach and SMEA versus NSGA-II. Besides, our algorithm ranks *first* in the Friedmans test(see Table 5 and Fig 16).

Table 6 represents the average value of C-metrics running 10 times independently on Examples 1–10. We used SMEA to make a pairwise comparison with three algorithms in the literature. We can found $C(SM, WS) \geq C(WS, SM)$, $C(SM, TE) \geq C(TE, SM)$ and $C(SM, NS) \geq C$

**Table 4. Results of the multiple problem Wilcoxon's test on the problems of F01-F10.**

| SMEA VS | $R^+$ | $R^-$ | p-value | $\alpha = 0.05$ |
|---|---|---|---|---|
| Weighted sum approach | 55 | 0 | 0.005 | Yes |
| Tchebycheff approach | 55 | 0 | 0.005 | Yes |
| NSGA-II | 45 | 10 | 0.044 | Yes |

**Table 5. Ranking of the SMEA by the Friedmans test on the problems of F01-F10.**

| Algorithm | Ranking |
|---|---|
| SMEA | **3.60** |
| Weighted sum approach | 1.70 |
| Tchebycheff approach | 1.90 |
| NSGA-II | 2.80 |

($NS$, $SM$) on problems 1, 2, 5, 6, 8, 10. It means that SMEA found a better non-dominated set than algorithms in [26].

In addition, CPU time is used to compare the efficiency of the algorithms. For the first 10 Examples, Table 7 shows the CPU times running 10 times independently used by both SMEA and two compared algorithms in [26]. From Table 7 one can see that SMEA can save more

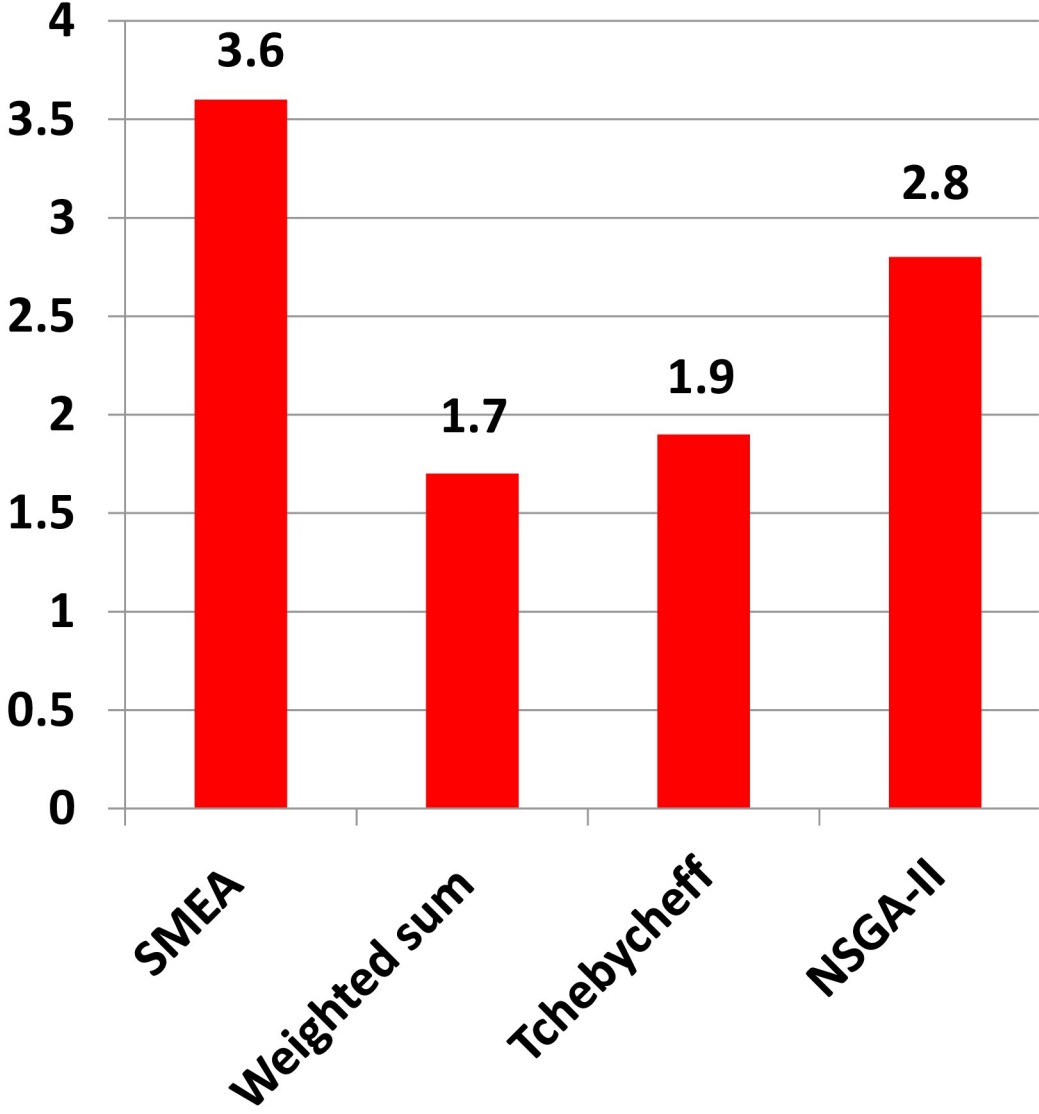

**Fig 16. Ranking of SMEA by Friedman test on the problems of F01-F10.**

**Table 6. Comparison of C-metric between SMEA and algorithms in [26].**

| Text problems | F01 | F02 | F03 | F04 | F05 | F06 | F07 | F08 | F09 | F10 | +/−/≈ |
|---|---|---|---|---|---|---|---|---|---|---|---|
| C(SM,WS) | **1.0000** | **0.6700** | 0.4700 | **0.5700** | **0.5400** | 0.0000 | 0.0200 | **0.0930** | 0.0067 | **0.8800** | 6/3/1 |
| C(WS,SM) | 0.0000 | 0.5700 | **1.0000** | 0.5100 | 0.4100 | 0.0000 | **0.3200** | 0.0067 | **0.8400** | 0.5400 | |
| C(SM,TE) | **1.0000** | **0.6700** | **0.5500** | 0.2600 | 0.0000 | 0.0000 | **0.0930** | **0.5200** | **0.6700** | **0.4100** | 7/1/2 |
| C(TE,SM) | 0.0000 | 0.5000 | 0.3900 | **0.5300** | 0.0000 | 0.0000 | 0.0800 | 0.4000 | 0.4800 | 0.0000 | |
| C(SM,NS) | **1.0000** | **0.6900** | **0.6200** | **0.3700** | **0.0067** | 0.0000 | 0.0013 | **0.0400** | **0.0470** | **0.9600** | 8/1/1 |
| C(NS,SM) | 0.0000 | 0.0130 | 0.0270 | 0.2900 | 0.0000 | 0.0000 | **0.0067** | 0.0200 | 0.0067 | 0.0000 | |

computation costs than the compared methods. For other Examples, we also provided their CPU times in Table 7.

It is difficult to solve BMPPs, especially to solve the follower's problem, and there exist only a small number of computational Examples. In the proposed algorithm, the weighted sum approach can transform the original multiobjective problem into single objective ones. Hence, SMEA can also be used to solve BLPPs with a single objective on which the effectiveness of the surrogate models can be verified. When SMEA is used in solving Examples 14–20, the best solution $(x^*, y^*)$ in all 10 runs is recorded as well as the corresponding objective values $F(x^*, y^*)$ and $f(x^*, y^*)$. All results are presented in Table 8 in which the objective values are denoted by Ref-$f(x^*, y^*)$ and Ref-$F(x^*, y^*)$, respectively.

Table 8 shows the average value of optimal results running 10 times independently on Examples 14–20. The best results obtained are highlighted bold in this table. Especially, the optimal solutions of Examples 14–16 and 18 are obviously better than those provided in the existing literature. It follows that the surrogate model technique is effective to solve the problems of this type.

**Table 7. CPU time used by the SMEA and the CPU time of the approaches in the literature [26].**

| Test problem | Weighted sum approach(s) | Tchebycheff approach(s) | SMEA approach(s) |
|---|---|---|---|
| F01 | 288.8705 | 269.1355 | **184.0437** |
| F02 | 290.4008 | 305.5410 | **279.4890** |
| F03 | 1153.4208 | 1169.1095 | **329.9828** |
| F04 | 1514.5671 | 1652.7553 | **579.3628** |
| F05 | 316.4801 | 381.0618 | **290.3375** |
| F06 | 344.4778 | 256.3112 | **238.1882** |
| F07 | 487.5152 | 579.9305 | **351.7422** |
| F08 | 608.9422 | 550.5114 | **453.6247** |
| F09 | 674.3858 | 691.0868 | **328.4653** |
| F10 | 1060.9020 | 1102.1321 | **837.5767** |
| F11 | — | — | **424.1344** |
| F12 | — | — | **385.6785** |
| F13 | — | — | **506.1157** |
| F14 | — | — | **12.5431** |
| F15 | — | — | **10.8668** |
| F16 | — | — | **11.7438** |
| F17 | — | — | **12.2328** |
| F18 | — | — | **6.8531** |
| F19 | — | — | **12.1533** |
| F20 | — | — | **8.5941** |

**Table 8. Comparison of the best results found by SMEA and the compared approaches.**

| Test problems | Ref-$f(x^*, y^*)$ | $f(x^*, y^*)$ | Ref-$F(x^*, y^*)$ | $F(x^*, y^*) \pm std$ | $(x^*, y^*)$ |
|---|---|---|---|---|---|
| F14 | −152.5005 | −152.2950 | −351.8333 | **-352.0990** ± 0 | $x^* = (0, 0, 0, 0, 0, 0, 0, 0, 0, 0)$<br>$y^* = (0, 0.6600, 8.8330, 0, 0, 0, 0, 0, 0, 0)$ |
| F15 | 57.4800 | 57.4800 | −6600.0000 | **-6648.1400** ± 0 | $x^* = (7.3600, 3.5500, 12.3500, 17.4600)$<br>$y^* = (0.9100, 10.0000, 29.0900, 0)$ |
| F16 | −1.0156 | −1.0156 | −18.6787 | **-18.6835** ± 0 | $x^* = (0, 2)$<br>$y^* = (1.8768, 0.9076)$ |
| F17 | 3.2000 | 3.2000 | −29.2000 | −29.2000 ± 0 | $x^* = (0.0001, 0.8999)$<br>$y^* = (0, 0.5999, 0.4001)$ |
| F18 | 7.6145 | 7.6148 | −1.2091 | **-1.2092** ± 0 | $x^* = 1.8885$<br>$y^* = (0.8892, 0)$ |
| F19 | 1.0000 | 1.0000 | 0.0000 | 0.0000 ± 0 | $x^* = (−0.8885, −0.1115, 0.7173, 0.2510, 0.1672, −0.6012, −0.5287, −0.7114, 0.3353, 0.9776)$<br>$y^* = (0.0797, 0.9856, 0.4052, 0 −0.9677, −0.5822, 0.6768, 0, 0.8700, −0.2090)$ |
| F20 | 100.0000 | 0.0000 | 0.0000 | 0.0000 ± 0 | $x^* = (0.9857, 28.6942)$<br>$y^* = (−19.0141, 8.6941)$ |

## Conclusion

BMPP is one of the known hardest optimisation models in knowing that this problem always accumulates computational complexity of both the hierarchical structure and multi-objective optimization. In order to reduce the computational cost of the problem, two efficient techniques are embedded in the proposed algorithm. One is the weighted sum method used to deal with multiple objectives in the follower's problem. The other is the surrogate model which can efficiently save the computational cost in obtaining bilevel feasible solutions. In addition, a heuristic crossover operator is designed by making use of the gradient information. The simulation results in 20 computational examples show the efficiency of the proposed algorithm.

## Supporting information

**S1 File.**
(XLS)

**S2 File.**
(XLS)

## Acknowledgments

Hong Li is thanked for useful discussions and supply some reference datas. We thank the editors and the anonymous reviewers for their professional and valuable suggestions.

## Author Contributions

**Data curation:** Hecheng Li, Hong Li.

**Formal analysis:** Hecheng Li.

**Methodology:** Yuhui Liu.

**Writing – original draft:** Yuhui Liu.

**Writing – review & editing:** Yuhui Liu.

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
