## [Decision Letter · Decision Letter 0]

17 Sep 2020

PONE-D-20-21853

An Evolutionary Algorithm Using Surrogate Models for Solving Bilevel Multi-objective Programming Problems

PLOS ONE

Dear Dr. Liu,

Thank you for submitting your manuscript to PLOS ONE. After careful consideration, we feel that it has merit but does not fully meet PLOS ONE’s publication criteria as it currently stands. Therefore, we invite you to submit a revised version of the manuscript that addresses the points raised during the review process.

We look forward to receiving your revised manuscript.

Kind regards,

Weinan Zhang

Academic Editor

PLOS ONE

Additional Editor Comments:

To read the reviews and the reading of the manuscript by myself. I decide to give a major revision for the current version of the manuscript.

'The research work was supported by the National Natural Science Foundation of China under Grant Nos. 61966030 and 11661067, the Natural Science Foundation of Qinghai Province under Grant No. 2018-ZJ-901 and the Key Laboratory of IoT of Qinghai under grant 2020-ZJ-Y16.'

'NO - Include this sentence at the end of your statement: The funders had no role in study design, data collection and analysis, decision to publish, or preparation of the manuscript '

Reviewers' comments:

Reviewer's Responses to Questions

**Comments to the Author**

1. Is the manuscript technically sound, and do the data support the conclusions?

Reviewer #1: Yes

Reviewer #2: Partly

2. Has the statistical analysis been performed appropriately and rigorously? 

Reviewer #1: Yes

Reviewer #2: N/A

3. Have the authors made all data underlying the findings in their manuscript fully available?

Reviewer #1: Yes

Reviewer #2: Yes

4. Is the manuscript presented in an intelligible fashion and written in standard English?

Reviewer #1: Yes

Reviewer #2: Yes

5. Review Comments to the Author

Reviewer #1: This paper proposed an evolutionary algorithm using surrogate models for solving bilevel multiobjective programming Problems. It is an interesting topic and can be accepted with revised. My detailed comments are as follows:

1) Figure 2 shows the framework of the entire algorithm, but in Figure 2 we can seen that the represents "Archive D2" it is unclear. And the color of the partial graph in Figure 15 is inconsistent with the color in other Figures (13-14), it is recommended to modify.

2) Please carefully check the representation of the numbers in Table 2 and Table 3. For example, did you miss "E" in the representation of numbers in Example F08-F10?

3) It can be seen from Table 6 that the C-Measure of different algorithms is compared, But the description of Table 6 on page 24 seems inappropriate, such as "… C(SM,WS) > C(WS,SM), C(SM,TE) > C(TE,SM) and C(SM,NS) > C(NS,SM) on problems 1,2,5,6,8,10… ", but I found that in F06, it should be " …C(SM,WS) = C(WS,SM), C(SM,TE) = C(TE,SM) and C(SM,NS) = C(NS,SM) …".

4) Table 8 shows the comparison of objective function values on Examples 14-20. Please clearly explain the relationship between F(x∗,y∗) and f(x∗,y∗). Whether the comparison of their values is mainly reflected in F(x∗,y∗)?

Reviewer #2: The paper presents a novel evolutionary algorithm for the bilevel multi-objective programming problem, which is a very difficult problem being studied extensively in previous studies. The experimental results demonstrate that the proposed method is very efficient for the problem. Overall, the paper needs major improvements in written, especially for the organization structure. It seems that the related work and the introduction are mixed together, which makes the reader difficult to understand the major motivation or how the idea is developed. In addition, the Section 5, the authors should pay more attention to let the algorithm being presented more understandable. The direct showing can make readers hard to follow.

6. PLOS authors have the option to publish the peer review history of their article (what does this mean?). If published, this will include your full peer review and any attached files.

Reviewer #1: No

Reviewer #2: No

---

## [Author Response · Author response to Decision Letter 0]

3 Nov 2020

We have revised and uploaded the file according to the reviewer’s comment

---

## [Decision Letter · Decision Letter 1]

1 Dec 2020

Evolutionary Algorithm Using Surrogate Models for Solving Bilevel Multiobjective Programming Problems

PONE-D-20-21853R1

Dear Dr. Li,

We’re pleased to inform you that your manuscript has been judged scientifically suitable for publication and will be formally accepted for publication once it meets all outstanding technical requirements.

Kind regards,

Weinan Zhang

Academic Editor

PLOS ONE

Additional Editor Comments (optional):

The main concerns of reviewers are addressed in the revision.

However, please also mind the comment pointed by the reviewer "The flowchart Fig.2 has no start and end marks." and revise accordingly.

Reviewers' comments:

Reviewer's Responses to Questions

**Comments to the Author**

1. If the authors have adequately addressed your comments raised in a previous round of review and you feel that this manuscript is now acceptable for publication, you may indicate that here to bypass the “Comments to the Author” section, enter your conflict of interest statement in the “Confidential to Editor” section, and submit your "Accept" recommendation.

Reviewer #1: (No Response)

Reviewer #2: All comments have been addressed

2. Is the manuscript technically sound, and do the data support the conclusions?

Reviewer #1: Yes

Reviewer #2: Yes

3. Has the statistical analysis been performed appropriately and rigorously? 

Reviewer #1: Yes

Reviewer #2: Yes

4. Have the authors made all data underlying the findings in their manuscript fully available?

Reviewer #1: Yes

Reviewer #2: Yes

5. Is the manuscript presented in an intelligible fashion and written in standard English?

Reviewer #1: Yes

Reviewer #2: Yes

6. Review Comments to the Author

Reviewer #1: In this study, the authors proposed an evolutionary algorithm using surrogate optimisation models to solve the computational complexity of both hierarchical structures and multiobjective optimisation in BMPP.

The authors have revised the paper according to the reviewer's suggestions.

The flowchart Fig.2 has no start and end marks.

Reviewer #2: The paper addresses an important problem and present a good solution. All previous issues have been handled. I think the paper is well done and suitable to be published directly.

7. PLOS authors have the option to publish the peer review history of their article (what does this mean?). If published, this will include your full peer review and any attached files.

Reviewer #1: No

Reviewer #2: No

---

## [Editor Report · Acceptance letter]

7 Dec 2020

PONE-D-20-21853R1 

Evolutionary Algorithm Using Surrogate Models for Solving Bilevel Multiobjective Programming Problems 

Dear Dr. Li:

I'm pleased to inform you that your manuscript has been deemed suitable for publication in PLOS ONE. Congratulations! Your manuscript is now with our production department. 

Kind regards, 

on behalf of

Dr. Weinan Zhang 

Academic Editor

PLOS ONE